# The potential for coral reef restoration to mitigate coastal flooding as sea levels rise

Lauren T. Toth [1] ✉, Curt D. Storlazzi [2], Ilsa B. Kuffner [1], Ellen Quataert[3], Johan Reyns [3,4], Robert McCall [3], Anastasios Stathakopoulos [1], Zandy Hillis-Starr[5], Nathaniel Hanna Holloway[5], Kristen A. Ewen[5], Clayton G. Pollock[5], Tessa Code[5] & Richard B. Aronson [6]

The ability of reefs to protect coastlines from storm-driven flooding hinges on their capacity to keep pace with sea-level rise. Here, we show how and whether coral restoration could achieve the often-cited goal of reversing the impacts of coral-reef degradation to preserve this essential function. We combined coral-growth measurements and carbonate-budget assessments of reef-accretion potential at Buck Island Reef, U.S. Virgin Islands, with hydrodynamic modeling to quantify future coastal flooding under various coral-restoration, sea-level rise, and storm scenarios. Our results provide guidance on how restoration of *Acropora palmata*, if successful, could mitigate the most extreme impacts of coastal flooding by reversing projected trajectories of reef erosion and allowing reefs to keep pace with the ~0.5 m of sea-level rise expected by 2100 with moderate carbon-emissions reductions. This highlights the potential long-term benefits of pursuing coral-reef restoration alongside climate-change mitigation to support the persistence of essential coral-reef ecosystem services.

Coral reefs provide critical barriers that can mitigate shoreline erosion and flooding for tropical coastal communities around the world[1–3]; however, climate change is jeopardizing the ability of reefs to continue to provide coastal protection by causing coral-reef degradation[4–6], sea-level rise[7–9], and changes in waves and storms[10,11]. Widespread losses of reef-building corals as a result of climate change and other anthropogenic impacts[12,13] have decreased reef elevation[14], reduced topographic complexity[15], and diminished the capacity for reef accretion to keep pace with sea-level rise[16]. Reefs whose vertical accretion lags rising sea level may no longer maintain their role as protective barriers, because dissipation of wave energy decreases as water depths increase[3,7].

The shallow reef-crest zone is responsible for the majority of wave dissipation over coral reefs and is, therefore, the most important habitat for shoreline protection[1,5,9]. The role of the reef crest is especially important in the western Atlantic, where the broad, shallow reef flats that help dissipate waves in the Indo-Pacific are generally absent[17]. Historically, reef crests in the western Atlantic were dominated by *Acropora palmata*, a species whose robust morphology and rapid growth made it the most important reef-builder in the region for hundreds of thousands of years[13,18,19]. Over the last half century, however, the combined impacts of coral disease and thermal stress have decimated *A. palmata* populations[12,13,18,20]. The unprecedented loss of this ecosystem engineer has fundamentally changed the structure and function of many western Atlantic reefs[13,18]; reef-crest habitats that once reached sea level are now significantly flatter and deeper in many locations[14,15].

For coral-reef-lined coasts, understanding the changing nature of the reef-building process, particularly in reef-crest habitats, is essential for estimating future risks from coastal hazards under sea-level rise[2,3]

[1]U.S. Geological Survey, St. Petersburg Coastal and Marine Science Center, St. Petersburg, FL, USA. [2]U.S. Geological Survey, Pacific Coastal and Marine Science Center, Santa Cruz, CA, USA. [3]Deltares, Delft, Netherlands. [4]IHE Delft Institute for Water Education, Delft, Netherlands. [5]National Park Service, Christiansted, VI, USA. [6]Florida Institute of Technology, Department of Ocean Engineering and Marine Sciences, Melbourne, FL, USA. ✉e-mail: ltoth@usgs.gov

and for developing effective coral-reef management strategies[13,21,22]. Many coral-restoration efforts are now reaching a scale at which they are expected to have positive impacts on reef function and ecosystem services such as shoreline protection[23]; however, the ability of coral restoration to achieve those broader goals has yet to be demonstrated.

Carbonate-budget studies provide a simplified model of the complex processes controlling the modern accretion-erosion balance and a means for estimating how reef accretion and, therefore, water depth, could change in response to the combined impacts of reef degradation and sea-level rise in the future[16,21]. Carbonate budgets can also be used to evaluate how management activities such as coral restoration could help reverse declines in reef building and support ecosystem function[22]; however, because of the difficulty of surveying the high-energy reef crest, few carbonate-budget studies have included this critical habitat[16]. Similarly, whereas hydrodynamic modeling studies have shown how increasing water depth over reefs can amplify coastal flooding in theory[5,7,8], including one recent analysis at our study area, Buck Island Reef National Monument (BIRNM)[24], to date no studies have combined these approaches to explore how ongoing changes in reef-accretion capacity could affect reefs' ability to provide coastal protection with and without management intervention.

We present a real-world example of the impacts of reef degradation on coastal flooding using data collected at BIRNM, located off the northeastern coast of St. Croix, U.S. Virgin Islands (Fig. 1), in the north-central Caribbean Sea[25]. Buck Island is surrounded by a narrow, emergent barrier reef approximately 10-m wide, enclosing a 1- to 4-m deep lagoon. Since the mid-1950s, shoreline erosion on the northwest side of Buck Island has reduced beach habitat[26] that has cultural and historical significance, supports the island's active tourism industry[27], and provides nesting habitat for endangered sea turtles[28]. Shoreline change coincided with dramatic degradation of reef ecosystems surrounding the island[26,29], but the potential contribution of reef degradation to shoreline erosion has yet to be evaluated.

Here, we first quantify spatial variability in reef-accretion potential —a measure of the maximum capacity for the vertical accretion of a reef—using census-based carbonate-budget models. We then use in situ measurements of *A. palmata* growth in combination with the carbonate-budget models to examine whether and how restoration of *A. palmata* onto the shallow reef crests at BIRNM could help close the gap between the reef and rising sea level to mitigate coastal flooding impacts in the future. Finally, we use hydrodynamic modeling to determine how projected changes in bathymetry from both reef erosion and restoration, in combination with sea-level rise, could impact coastal flooding potential during storms by 2100. We demonstrate that although sea-level rise will be the primary driver of coastal flooding, reef erosion will result in a small, but measurable increase in flooding potential by the end of the century. Our study also provides insight into how successful coral-reef restoration efforts initiated within the present decade could, in combination with larger-scale climate-change mitigation, reduce the impacts of sea-level rise and wave-driven water levels on coral-reef-lined coasts like BIRNM.

## Results

Based on our carbonate budget reef census in 2016, reef-accretion potential varied between -5.31 and 6.21 mm y$^{-1}$ and averaged -1.56 mm y$^{-1}$ ($\pm$0.27 standard error [SE]) across 54 sites within shallow fore-reef, reef-crest, and back-reef habitats ($n = 18$ sites each; Fig. S1) on the southern and northern sectors ($n = 27$ sites each) around BIRNM (Fig. 1a; Tables S1 & S2). Only five sites had positive reef-accretion potential, and, on average, reef-accretion potential was only positive on the southern reef crest (0.10 mm y$^{-1}$ $\pm$ 1.20 SE versus $\leq$-1.41 mm y$^{-1}$ in all other zones). As a result, reef-accretion potential in the southern sector was significantly higher than in the northern sector (Linear mixed-effects model [LME]$_{sector}$: $F_{1,16} = 5.72$, $p = 0.03$; Tukey test: $p < 0.05$). Average projected elevation change of the southern reef crest by 2100 was negligible at +1 cm ($\pm$10 cm). In contrast, bioerosion was the dominant process at 91% of sites, with estimated decreases in elevation due to reef erosion averaging -13.09 ($\pm$0.02) cm by 2100 (Tables S1 & S2).

Estimated bioerosion, which was primarily driven by the parrotfish *Sparisoma viride* (Table S3), did not vary significantly among reef sectors or habitats (Fig. 1b; LMEs: $p > 0.05$). The relatively higher reef-

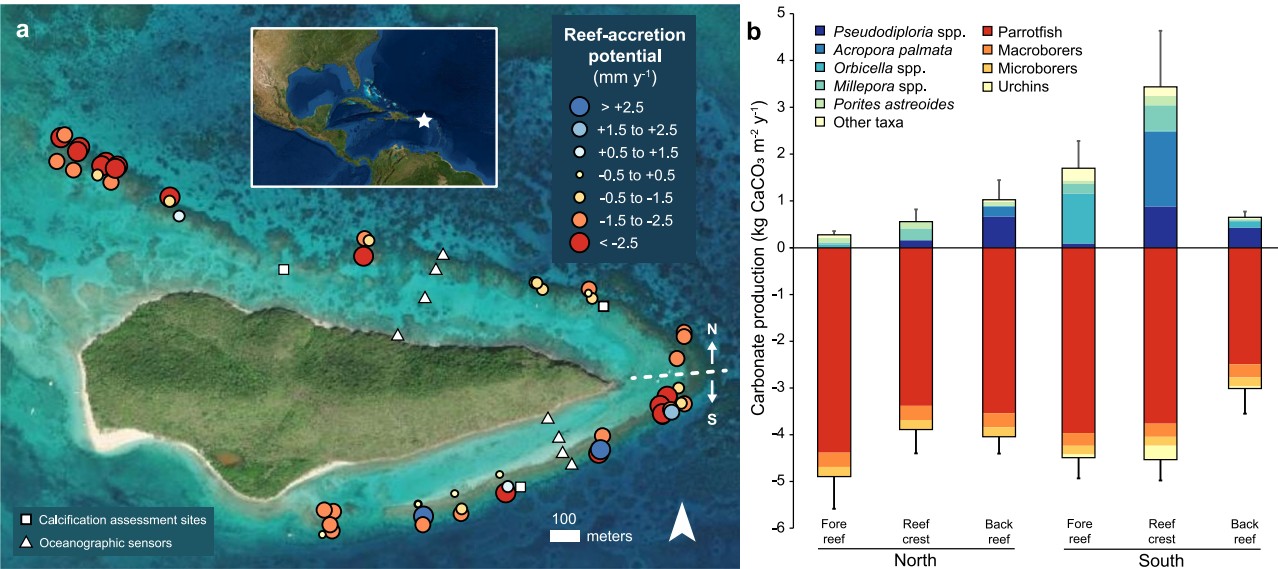

**Fig. 1 | Map of the study area and carbonate budgets at Buck Island Reef National Monument in 2016, highlighting the significantly higher carbonate production and coral reef-accretion potential in the southern reef crest compared with the other reef zones. a** Estimated mean reef-accretion potential (mm y$^{-1}$) at 54 sites from the fore-reef, reef-crest, and back-reef habitats around the island. Locations where in situ growth of *Acropora palmata* and *Pseudodiploria strigosa* were quantified (white squares) and transects where in situ tide and wave sensors were deployed for calibration and validation of the hydrodynamic model (Fig. S4) are also shown **b.** Bar plot showing mean ($\pm$ standard error) gross carbonate production by calcifying reef taxa (upper plot) and bioerosion by bioeroding groups (lower plot) across reef zones. Map image is the intellectual property of Esri and is used herein under license. Copyright 2020 Esri and its licensors. All rights reserved.

accretion potential on the southern reef crest was instead a result of significantly higher coral cover (Table S1; 14.88% ± 4.05 on the southern reef crest versus <7% in the other reef zones, on average), which resulted in higher gross carbonate production there (Fig. 1b; 3.43 ± 1.20 kg CaCO$_3$ m$^{-2}$ y$^{-1}$ on the southern reef crest versus ≤1.70 kg CaCO$_3$ m$^{-2}$ y$^{-1}$ in the other zones, on average; LMEs: $F_{1,16}$ = 7.58, $p$ = 0.01 and $F_{2,32}$ = 3.92, $p$ = 0.03 for sector and the sector and habitat interaction, respectively; Tukey tests: $p < 0.05$ for the southern reef crest vs. the northern reef-crest, northern fore-reef, and southern back-reef habitats). *Pseudodiploria* spp., *A. palmata*, *Orbicella* spp., *Millepora* spp., and *Porites astreoides* were the most significant contributors to carbonate production at BIRNM (Fig. 1b); all other taxa contributed <0.05 kg CaCO$_3$ m$^{-2}$ y$^{-1}$ on average in any reef zone. Differences in the abundances of those taxa explain the variability in carbonate production among reef zones (ANOSIM: $R$ = 0.24, $p$ = 0.001), with higher production by *A. palmata*, *Pseudodiploria* spp., and *Millepora* spp. on the southern reef crest (SIMPER analysis; Table S4).

## Potential impacts of restoring reef-building corals

During the coral-growth experiment we conducted at BIRMN from 2019 to 2021, calcification rates of *A. palmata* ranged from 21.04 to 42.91 kg m$^{-2}$ y$^{-1}$, with an average of 29.07 kg m$^{-2}$ y$^{-1}$ (±1.26). The average rate of change in the height of *A. palmata* colonies was 6.99 cm y$^{-1}$ (±1.56; range = 4.19–9.25 cm y$^{-1}$), with average planar surface area of the colonies increasing by 196.21 cm$^2$ y$^{-1}$ (±102.95; range = 51.67–407.80 cm$^2$ y$^{-1}$). Although 14 of the 30 *A. palmata* colonies included in the experiment died within days of transplantation due to the occurrence of an early thermal-stress event at BIRNM during the experimental set-up, only one colony died in the subsequent two years of the experiment.

Calcification rates of *Ps. strigosa* colonies in our experiment ranged from 7.96 to 21.31 kg m$^{-2}$ y$^{-1}$, with an average of 14.30 kg m$^{-2}$ y$^{-1}$ (±0.86). The average annual change in height of *Ps. strigosa* was 0.95 cm y$^{-1}$ (±0.71; range = -0.50 to 2.70 cm y$^{-1}$) and its change in planar surface area averaged 12.88 cm$^2$ y$^{-1}$ (±9.71; range = -5.10 to 35.45 cm$^2$ y$^{-1}$). Survival of *Ps. strigosa* was high throughout the experiment with only four colonies experiencing complete mortality and an additional four experiencing partial mortality. Stony coral tissue-loss disease (SCTLD) was just starting to impact some colonies during the last time interval (see photographs in[30]). Overall, although there was some variability in coral growth across space and time (Table S5), our results indicate that both species of reef-building corals can grow rapidly throughout BIRNM (Fig. 2a).

To evaluate how coral restoration could harness the rapid growth of *A. palmata* to improve reef state and function, we first developed a simple population model (assuming no reproduction) based on average, planar growth rates of *A. palmata* from our study and literature-derived mortality and fragmentation rates to project potential changes in *A. palmata* cover on the BIRNM reef crest (Fig. S1) under various restoration and mortality scenarios (see Methods). Our models indicate that a single, large-scale *A. palmata* outplanting effort before 2030 could result in increases in coral cover on the reef crest at BIRNM for the next 20 years, with restoration projected to increase *A. palmata* cover by 8.5%, 21.2%, and 42.4% under our fixed-mortality scenario (i.e., 60% net survival per decade; see Methods) following outplanting of 100,000 (0.5 m$^{-2}$), 250,000 (1.5 m$^{-2}$), and 500,000 (3 m$^{-2}$) colonies, respectively (Fig. 2b). After 2050, without additional outplanting or the establishment of a reproductive, self-sustaining *A. palmata* population, continuing mortality would result in a gradual diminution of coral cover. The most rapid declines would occur under the scenario of increasing climate-related mortality (i.e., a 5% increase in mortality per decade [see Methods]; dashed lines in Fig. 2b). A more gradual decline would occur if climate-related mortality moderates over time (i.e., a 5% decrease in mortality per decade through increased acclimatization, adaptation, or resilience [see Methods];

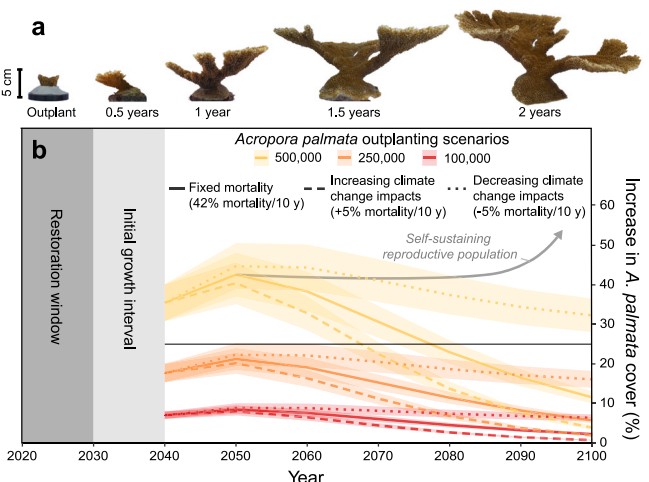

**Fig. 2 | Illustration of the rapid growth of *Acropora palmata* and its potential for increasing coral cover through restoration. a** Example photographic time-series showing the growth of a colony of *A. palmata* at Buck Island Reef National Monument (BIRNM) over two years[30]. **b** Modeled mean (± standard error) increases in the percent cover of *A. palmata* on the BIRNM reef crest following three scenarios of reef-crest restoration until 2030 and three scenarios of post-restoration mortality (see Methods). The model indicates that a single large-scale restoration effort now could allow reefs at BIRNM to maintain high levels of coral cover until around 2050; however, additional local management, global reduction of carbon emissions, and/or the recovery of ecological processes like sexual reproduction (illustrated conceptually by the gray arrow) or a decrease in climate-related impacts, potentially through acclimatization and/or increased resilience (dotted lines), would be necessary to maintain stable coral populations in the long term. The horizontal line in **b** denotes the minimum increase in *A. palmata* cover (25%, to ≥31% total cover) our carbonate budgets suggest is needed for reef-accretion potential to keep pace with lowest projections of sea-level rise for 2100 (Fig. 3).

dotted lines in Fig. 2b). Overall, the model indicates that by the end of the century, the effects of restoration in enhancing *A. palmata* cover would largely be reversed under most of the modeled scenarios (i.e., <10% net increase in coral cover); however, in more optimistic scenarios (i.e., fixed mortality with 500,000 initial outplants and reduced mortality with 250,000 initial outplants), *A. palmata* cover is projected to remain elevated by more than 10% to 2100. The best outcome in our model was for the reduced-mortality scenario with 500,000 initial outplants, for which a >30% increase in *A. palmata* cover was projected to persist to 2100.

In this most optimistic scenario, a sustained increase of 30% *A. palmata* cover above the 2016 baseline coral cover of 6.14% ± 1.40 and 14.86% ± 4.03 (0.01 ± 0.01 and 4.42 ± 2.37% *A. palmata* cover) in the northern and southern sectors of the BIRMN reef crest, respectively, would bring total coral cover in those zones to ~36 and 45%. Under that scenario, our carbonate-budget models indicate that restoration could increase reef-accretion potential on the BIRNM reef crest to 7.38 mm y$^{-1}$ (±0.60) on average (6.59 ± 0.70 and 8.16 ± 0.94 mm y$^{-1}$ for the northern and southern sectors, respectively). At that rate, the reef crest could be capable of as much as 0.25 m (±0.02) of vertical accretion on average by 2050 and 0.62 m (±0.05) of vertical accretion by 2100 (0.55 ± 0.06 and 0.68 ± 0.08 by 2100 for the northern and southern sectors, respectively), which would be sufficient to allow the reef crest throughout BIRNM to keep pace with Low to Intermediate-Low predictions of sea-level rise for the mid- and end-of-century[31] (Fig. 3). In contrast, although the reef crest is projected to have positive reef-accretion potential with an increase of at least 10% *A. palmata* cover, none of the restoration scenarios with less than a 25% increase in *A. palmata* cover were estimated to be sufficient for the reef to keep pace with even the most conservative projections of sea-level rise.

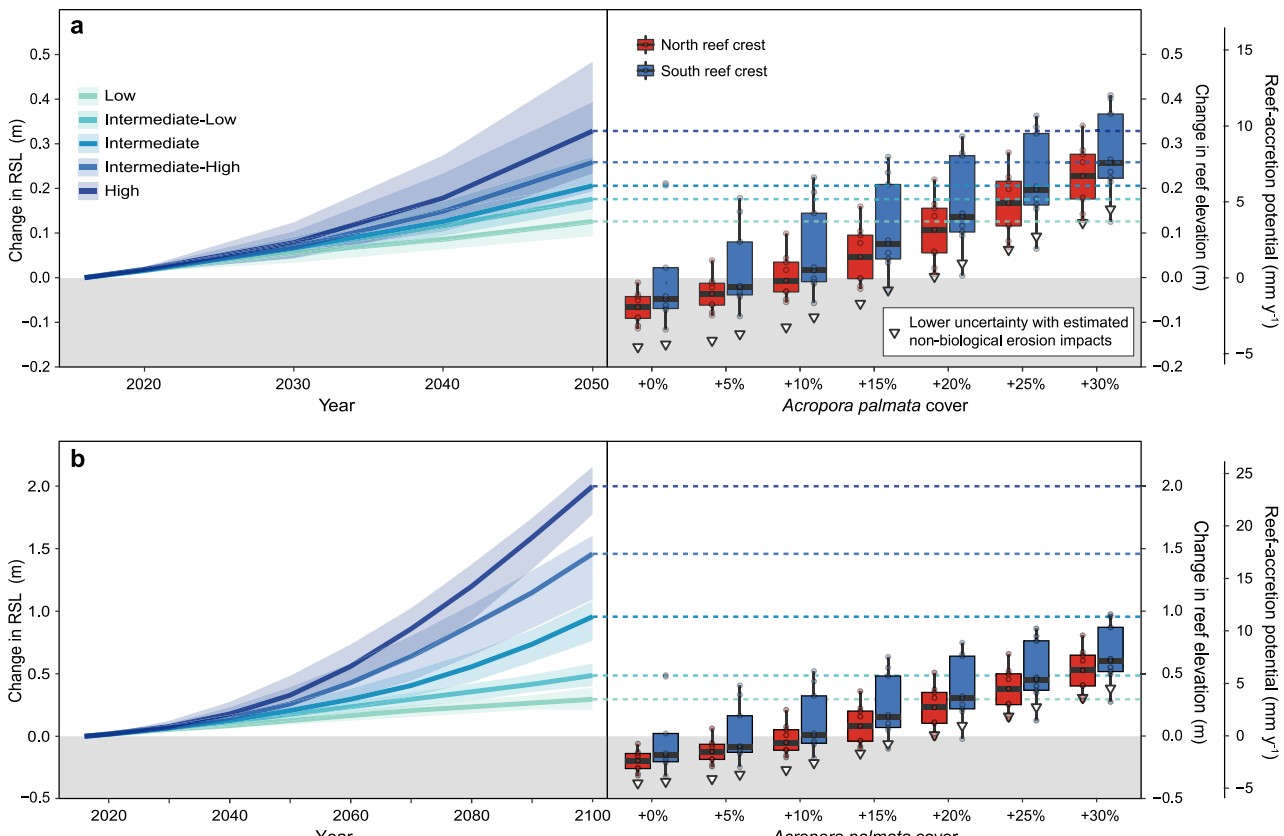

**Fig. 3 | Carbonate-budget models showing how increases in *Acropora palmata* through restoration could improve the ability of reef-crest habitats at Buck Island Reef National Monument (BIRNM) to keep pace with future sea-level rise.** These results indicate that if *A. palmata* cover could be increased by ~30% (to total mean cover of 36 and 45% for the northern and southern reef crest, respectively), the reef could keep pace with Low to Intermediate-Low projections of relative sea-level (RSL) rise. Median projections (solid to dashed lines; shading represents two-standard-deviation confidence intervals) of future relative sea-level rise for St. Croix (relative to 2016; NOAA Tide Station # 9751401; ref. 31) to 2050 **a** and 2100 **b** are based on the five (Low to High) sea-level-rise scenarios evaluated in Sweet et al.[31]. The sea-level-rise projections (left) are compared with carbonate-budget-based projections of median reef-elevation change (solid black horizontal lines) and reef-accretion potential (right; boxplots) of BIRNM reef-crest habitats based on our surveys in 2016 (+0%) and under restoration scenarios of 5–30% increases in *A. palmata* cover. Boxes bound the first and third quartiles and whiskers represent 1.5× the inter-quartile ranges. White triangles represent a lower uncertainty on median projected elevation changes based on an estimate of potential erosion processes not accounted for in the carbonate budgets (see Methods). Note that these uncertainties are generally encompassed by the 1.5× inter-quartile range of the boxplots in the more optimistic restoration scenarios. Areas representing negative elevation change and reef-accretion potential are shaded in gray.

## Coastal protection during storms

Modeled total water levels—which include the influence of tides, waves, and storm surge—increase non-linearly with rising sea level and more extreme wave conditions because sea-level rise allows for larger waves to propagate across the reefs (Figs. 4 and 5). With present-day bathymetry, this wave-driven component of the maximum total water levels averaged 4.17 m (±0.03) under the 10-year storm scenario (i.e., a tropical storm or Category-1 hurricane[32]: a "minor" storm) and 4.94 m (±0.02) under the 50-year storm scenario (i.e., a Category-5 hurricane[32]: a "major" storm). Maximum total water levels vary along the shoreline due to variability in nearshore bathymetry and its subsequent effects on waves and wave-driven water levels; however, the alongshore variation declines as sea level increases, because greater water depth results in relatively lower bathymetric variability on the reef that, in turn, causes lower frictional wave dissipation. The overall trends were similar when projected changes in reef elevation were incorporated into the model (Fig. S2); however, maximum total water levels reaching the shoreline were consistently higher in the northern sector where the elevation of the reef is projected to erode by almost 0.2 m on average by 2100 (Table S1), compared with the southern sector where the reef crest is projected to maintain its present-day elevation (Fig. 4c, d).

Greater water depths due to sea-level rise allow greater wave energy to propagate across the reefs, resulting in larger sea-swell waves (i.e., wave periods of 5–25 s) nearshore. Whereas the mean infragravity wave (i.e., wave periods >25 s) heights are less affected by sea-level rise, maximum values are more sensitive (Fig. S3). Together, these wave components contribute to an approximately linear increase in wave-driven water levels with sea-level rise that, combined with sea-level rise itself, results in a non-linear increase in total water levels and thus coastal flooding potential.

Because substantial coral-restoration efforts (i.e., increasing 30% *A. palmata* cover to ≥36% total cover) could promote reef accretion and help narrow the gap between reef elevation and sea level by more than 0.5 m on average by 2100 (Fig. 3), restoration could substantially reduce total water levels during future storms (Fig. 5b). Restoration therefore has the potential to negate the wave-driven component associated with low-end sea-level-rise scenarios (i.e., by keeping pace with a rise of 0.2–0.5 m by 2100) and to substantially reduce total water levels, by up to 0.81 m (±0.03) on average, under intermediate-to-high sea-level rise scenarios (i.e., 1.2–2.0 m by 2100; Fig. 5b). Furthermore, by mitigating wave-driven increases in total water levels, successful, large-scale restoration could essentially reduce the flooding impacts of a major hurricane (50-year storm) to

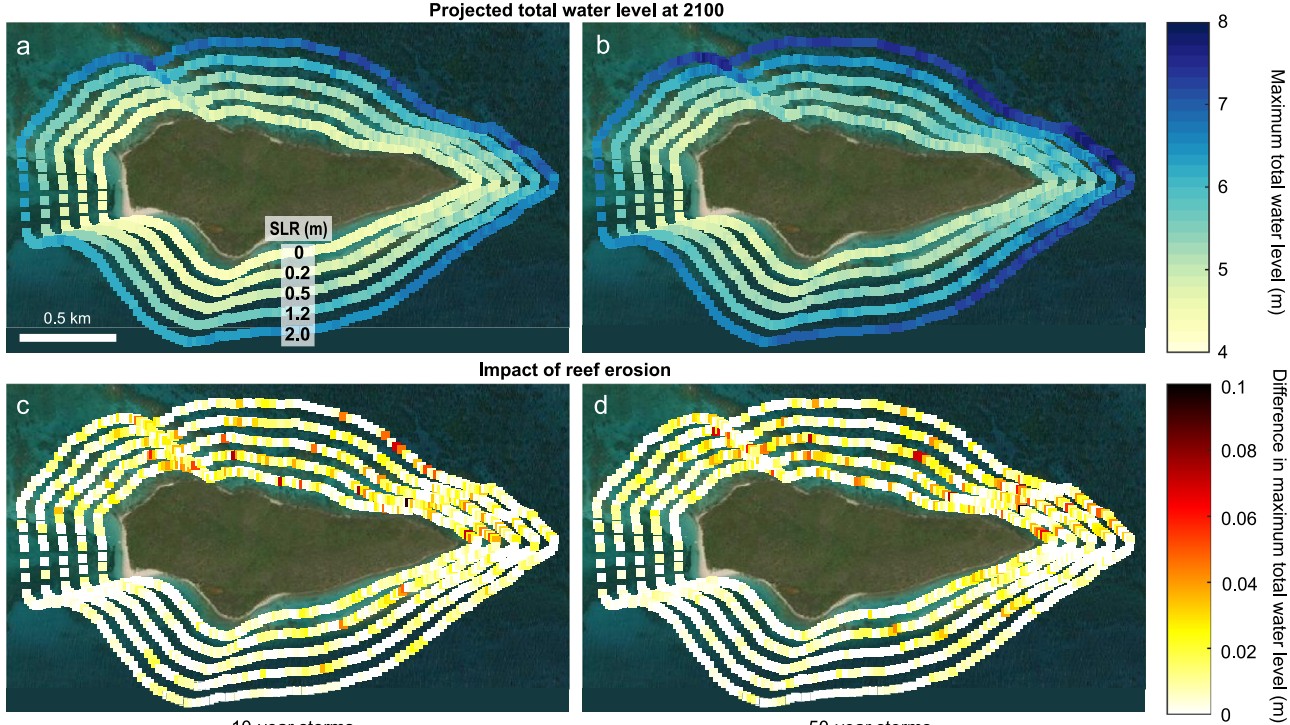

**Fig. 4 | Maps displaying the impact of sea-level rise and coral-reef bathymetry on alongshore variation in total water levels and thus coastal flooding potential.** In each panel, the maximum total water levels are plotted along the shoreline of Buck Island, with each ring of points representing a sea-level rise scenario, from +0.0 m (inner ring) to +2.0 m (outer ring). Left (**a, c**) and right (**b, d**) columns present results for 10-year storm (i.e., a strong tropical storm or Category-1 hurricane[32]) and 50-year storm scenarios (i.e., a Category-5 hurricane[32]), respectively. The top row (**a, b**) provides projections of maximum total water levels at 2100 with projected reef erosion from the carbonate-budget models under the different storm and sea-level rise scenarios. The bottom row (**c, d**) illustrates the impact of reef erosion by showing the difference in projections of maximum total water levels between models run with projected reef erosion (**a, b**) and those run with present-day bathymetries (Fig. S2). The plots indicate that although sea-level will be the most significant driver of increases in total water levels and thus flooding potential, higher total water levels under all sea-level-rise scenarios are projected for the northern sector of the island, corresponding to the areas of greatest projected reef erosion in Fig. 1. Map image is the intellectual property of Esri and is used herein under license. Copyright 2020 Esri and its licensors. All rights reserved.

that equivalent to a strong tropical storm or minor hurricane (10-year storm; Fig. 5b).

## Discussion

The capacity of coral reefs to keep pace with sea-level rise is central to their ability to continue to provide shoreline protection to coastal communities in the future[3,5,7–9,16]. For millennia, rapid accretion of reefs dominated by the reef-crest ecosystem engineer *A. palmata* has supported this key reef function in the western Atlantic[13]. Beginning in the late 1970s, however, white-band disease caused dramatic declines of acroporid populations throughout the region, including at BIRNM[20,29,33,34]. Whereas in 1976 cover of *A. palmata* was >30% on average, by the late 1980s, average *A. palmata* cover had been reduced to <2%[29,34]. Coral bleaching and associated disease outbreaks in 2005 further diminished the remaining populations of reef-building corals at BIRNM and there has been limited recovery since[25]. Gladfelter et al.[35] estimated that in the late 1970s, carbonate production by *A. palmata* at BIRNM was as high as 15.18 kg CaCO$_3$ m$^{-2}$ y$^{-1}$, but our study indicates that average production by all species combined is now an order of magnitude lower at just 1.27 (±0.27) kg CaCO$_3$ m$^{-2}$ y$^{-1}$. Bythell et al.[29] hypothesized that the loss of *A. palmata* may have been sufficient to halt reef accretion at BIRNM, and our study supports that conclusion.

We found that by the time of our surveys in 2016, not only was reef-accretion potential at BIRNM well below the average regional Holocene baseline accretion rate of 3.09 mm y$^{-1}$ (refs. 34,36; cf. 37,38) and the contemporary western Atlantic reef-accretion potential average of 1.87 mm y$^{-1}$ (ref. 16), erosion had become the dominant process on most of the reef, with reef-accretion potential averaging -1.56 mm y$^{-1}$. This rate is only slightly lower than the erosion rate of -2.7 mm y$^{-1}$ that Yates et al.[14] estimated from landscape-scale changes in seafloor elevation around Buck Island from 1981–2014, and is comparable to the highest rates of net erosion estimated for other reefs in the western Atlantic[16,22,39,40]. The few sites in our study that had positive carbonate budgets all had coral cover >16% and were dominated by *A. palmata* and/or the massive reef-building corals *Ps. strigosa* and *Orbicella* spp. (Fig. 1b; Table S4), highlighting the importance of key reef-building corals in maintaining positive reef-accretion potential[22,41,42]. Our somewhat higher coral-cover threshold for positive reef-accretion potential compared with the 10% Caribbean-wide threshold[38] is likely a result of the 2.5-times higher estimated contribution of bioerosion at BIRNM (-4.14 kg CaCO$_3$ m$^{-2}$ y$^{-1}$ vs. Caribbean-wide average of -1.64 kg CaCO$_3$ m$^{-2}$ y$^{-1}$; ref. 16). This difference is primarily due to the relatively high rate of parrotfish bioerosion estimated in our study, as well as the higher macrobioerosion rates incorporated in our budgets based on in situ measurements of macrobioerosion by Whitcher[43] (cf.[16,37]). It is possible that parrotfish bioerosion was overestimated in our study because our carbonate budgets incorporated the relatively high parrotfish bioerosion rates suggested by the ReefBudget v1 methodology[44] (see Methods); however, BIRNM is also one of the oldest no-take marine reserves in the western Atlantic (established in 1961), which has resulted in elevated parrotfish biomass there compared with nearby locations open to fishing[45]. Because many parrotfish scrape and excavate the reef structure as they graze, protection of these key herbivores could have the unintended consequence of elevating reef bioerosion[12].

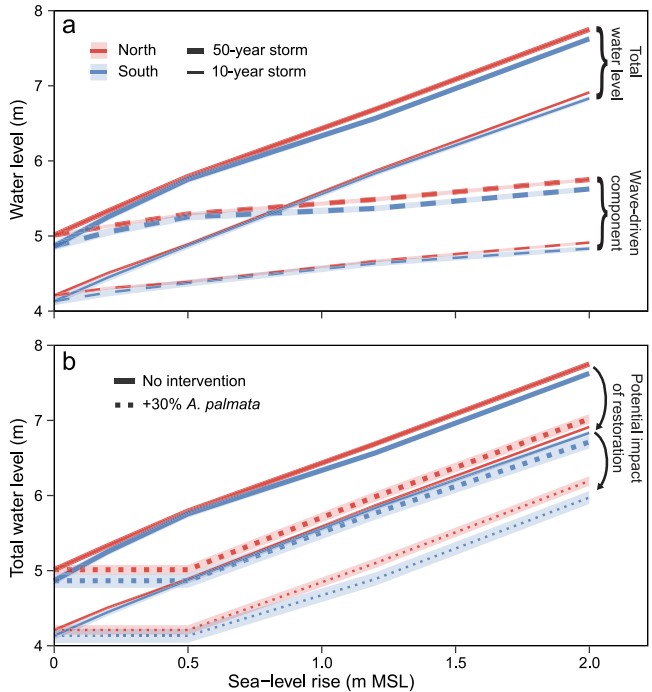

**Fig. 5 | Summary of the alongshore mean (± standard error [SE]) of maximum total water levels predicted to reach modeled sites (shown in Fig. 4) along the northern (*n* = 145) and southern (*n* = 71) shorelines of Buck Island by 2100 under sea-level rise (+0 to +2.0 m), storms (10- and 50-year storms), and coral restoration scenarios. a** Mean (±SE) maximum total water levels for each sea-level rise and storm scenario with reef degradation (solid lines) and the wave-driven component of total water levels (dashed lines). **b** Comparison of the total water levels with reef degradation in **a** to mean (± SE) total water levels estimated with reef elevation changes (Fig. 3) projected under a 30% increase in *A. palmata* cover through restoration (dotted lines), demonstrating how successful restoration could help mitigate some of the impacts of future sea-level rise at Buck Island (i.e., by reducing total water levels by up to 0.81 m). Note that because this figure presents spatially averaged trends, the alongshore variability apparent in Fig. 4, including areas of non-linear total-water-level changes, is less clear. Uncertainties (SEs) for the restoration scenario are the root-sum-squares of the uncertainties in total water levels and of projected reef-elevation change with restoration.

Overall, we found that only one habitat—the southern reef crest—exhibited marginally positive reef-accretion potential, as a result of the relatively high abundances of *A. palmata* and *Ps. strigosa* there (Fig. 1b). Whereas most reef zones at BIRNM are projected to lose 0.12–0.23 m of elevation on average by 2100, the southern reef crest is predicted to more-or-less maintain its present-day elevation (Table S2). We found that although rising sea levels will be the dominant driver of future coastal-flooding risk at BIRNM[24], the impacts of sea-level rise will be amplified by the diverging patterns of geomorphic change, with higher maximum total water levels predicted in the northern sector where the reef is eroding across all sea-level and storm scenarios (Fig. 4c, d; cf.[5,7,8,24]). This difference would cause alongshore gradients in waves and wave-driven water levels at the east and west ends of the island, likely resulting in net erosion and sediment transport to the south[46]. Indeed, since 1954, the primary beach habitat on the western side of the island—which supports important cultural and ecological resources such as critical nesting habitat for endangered sea turtles[26,28]—has migrated to the south, and nearly 30% (0.88 ha) of the total beach area has been lost due to storm-driven erosion[26]. For coral-reef-lined coasts fronted with wider shallow-water reefs than the bank-barrier reef at BIRNM, reef erosion would have an even greater impact on the flooding potential[5,47].

Although our results indicate that the southern reef crest at BIRNM was maintaining its role as a coastal barrier at the time of our surveys in 2016, its future role in providing coastal protection is precarious given the inevitability of continuing coral loss from climate change and other episodic natural and human impacts[12]. Indeed, just one year after our surveys were completed, Hurricane Maria passed ~60 km west of BIRNM as a Category-5 storm and caused significant damage to southern reef habitats. Many of the large *A. palmata* colonies that were responsible for the southern reef crest's rapid rate of carbonate production (Fig. 1b) were toppled and fragmented[25], although many were later restabilized by Park staff. This disturbance highlights the importance of physical erosion from storms—a process not considered explicitly in our carbonate budgets (but see below)—in modulating long-term reef accretion[16,22,48]. Most recently, in 2021, SCTLD began to impact the reef at BIRNM[25,30]. Acroporids are not susceptible to SCTLD, but the disease has had significant impacts on *Pseudodiploria* and *Orbicella* spp. (https://floridadep.gov/sites/default/files/Copy%20of%20StonyCoralTissueLossDisease_CaseDefinition%20final%2010022018.pdf), which were dominant in many reef habitats at the time of our surveys (Fig. 1b) and historically[25,29,35]. Together, the omission of the important impacts of physical erosion from our carbonate budgets and the recent declines in coral populations suggest that our estimates of present-day reef-accretion potential are overly optimistic, and erosion is likely already more significant than our study suggests.

Increasing investments in expanding the scale of coral restoration in recent years[23] present an opportunity to begin to return reefs like BIRNM closer to more resilient historic baselines. Many restoration programs also now cite maintenance of key reef functions, such as shoreline protection, among their broader goals[23]; however, ours is one of the few studies to assess how and whether those functional impacts could be achieved. With high survival and average vertical growth rates of ~70 mm y$^{-1}$ at BIRNM, our results indicate that individual outplanted *A. palmata* colonies could grow as fast as sea-level rise even under the most pessimistic scenarios predicted by 2100 (i.e., up to 13.4 mm y$^{-1}$ in St. Croix[31]). By increasing reef complexity and minimizing the divergence between the elevation of the reef crest and sea level, restoration of a living barrier of *A. palmata* colonies could have the near-term effect of increasing wave breaking across the reef crest[3,6,7]. This could prevent additional wave-driven increases in water levels, and, therefore, help mitigate coastal flooding[49]. In the longer-term, the ability of restoration to continue to support coastal protection will depend on its impact not just on coral growth, but on the process of reef accretion.

Whereas coral restoration is often seen as simply a short-term stop-gap strategy to buy time until the larger threat of climate change can be addressed[23], our study provides important guidance on what it would take for short-term restoration efforts now to have a sustained impact on coastal protection in the long-term. Our models indicate that, if successful, a significant effort to restore *A. palmata* populations at BIRNM in the near-term (1.5–3.0 individuals m$^{-2}$ by 2030) could result in substantial and sustained increases in coral cover and, therefore, reef-accretion potential until the end of the century, even in the absence of sexual reproduction (Figs. 2 and 3); however, our results also support the conclusion that under a regime of continuing coral mortality from disease outbreaks, bleaching, and various local stressors[12], maintaining functional impacts of such one-off restoration activities will likely require some increase in coral resilience, acclimatization, and/or adaptation[23,50]. Under accelerating or even present-day mortality rates, additional restoration efforts would be needed to maintain elevated coral cover long-term. Fortunately, recent studies have demonstrated that even today's diminished coral populations still have the genetic diversity to support increased climate resilience[50]. Ideally, restoration would result in the reestablishment of a sexually reproductive, self-sustaining population[51] (dashed line in

Fig. 2b), which would increase both coral cover and genetic diversity and thus the capacity for adaptation[50], especially as historic population declines have limited sexual reproduction and favored asexual fragmentation in *A. palmata* in recent decades[52].

If the most optimistic restoration scenario, in which *A. palmata* cover is increased by ~30% (to ≥36% total cover), can be realized, our carbonate-budget models indicate that reef-accretion potential of the reef crest at BIRNM could exceed 5 mm y$^{-1}$ (Fig. 3). That rate is comparable to average regional Holocene baselines of reef accretion[36], but lower than the >10 mm y$^{-1}$ maximum Holocene accretion rates by *A. palmata* reefs[36]. It is also too slow for increases in reef elevation to meet the maximum total sea-level change projected by 2100 (i.e., ~2 m in the High scenario[31]), which could occur under the highest $CO_2$ emissions scenarios[31] (i.e., shared socioeconomic pathway [SSP]5-8.5). At present, however, the most probable magnitude of end-of-century sea-level rise across all $CO_2$-emission scenarios is between ~0.5 and 1 m (i.e., the Intermediate-Low to Intermediate scenarios in[31]; Fig. 3), with a higher probability of limiting sea-level rise to ~0.5 m if moderate mid-century reductions of $CO_2$ emissions are achieved (i.e., SSP1-2.6 or SSP2-4.5)[31]. Our models indicate that under that scenario significant and successful restoration (i.e., +30% *A. palmata* cover) could allow the BIRNM reef crest to keep pace with sea-level rise to 2100 (i.e., ~0.5 m; Fig. 3). This outcome would reduce total water levels during storms by >10% (0.81 m) under higher sea-level-rise scenarios (Fig. 5) and thereby decrease the potential flooding associated with major (Category-5) hurricanes to levels projected for more minor (Category-1 or tropical storms) storms with no mitigation. Given the prediction that climate change will increase the frequency of severe storms[11], the ability of coral reefs to mitigate the worst impacts of storms will become increasingly important in the future.

The specific projections from our model are only directly applicable to the baseline ecological and geomorphic setting of BIRNM; however, our overall conclusion—that the worst effects of sea-level rise on coastal flooding could be mitigated if restoration of reef-building species like *A. palmata* can return reef-accretion rates to historic baselines—is relevant for coral reefs throughout the western Atlantic. In fact, given relatively high bioerosion rates we estimated for BIRNM, the level of coral cover needed for reef-accretion potential to match projected rates of sea-level rise may be lower in many other western Atlantic locations[16,38]. On the other hand, reefs that have already fallen behind present day sea level[14] would have a higher deficit to make up. Developing location-specific coral-restoration strategies is essential for quantifying how broader management goals like shoreline protection can be met[23,53] and our study demonstrates the utility of carbonate-budget models as a starting point for assessing those types of functional metrics.

Our carbonate-budget and coral-growth models necessarily present a simplified version of reality that cannot consider the full complexity of reef ecology or the full suite of processes affecting reef accretion and erosion now and in the future[48,54]. Reef-accretion potential represents a high-end estimate of realized long-term reef accretion because carbonate budgets typically do not consider event-driven physical erosion or chemical dissolution[16,48]. Hurricanes in particular have been shown to have substantial and immediate impacts on Caribbean coral populations and reef struture[55,56]; however, storm-generated rubble also often contributes significantly to reef accretion[34] and storm-driven fragmentation is an essential and even beneficial component of the life history of branching corals like *A. palmata*[56–58]. The severe and immediate short-term impacts of storms may, therefore, translate to more moderate impacts on long-term accretion under present-day storm regimes; however, as climate change increases the frequency of severe storms[11], their effects on reef erosion could likewise increase.

To estimate how the omission of non-biological erosion could affect our results, we followed the approach of Toth et al.[22] and

calculated a lower uncertainty on reef-accretion potential based on the difference between maximum Holocene reef accretion by *A. palmata* in St. Croix of 13.4 mm y$^{-1}$ (ref. 36) and estimated historic reef-accretion potential based on the maximum observed *A. palmata* cover there in the 1970s of 62%[35] (see Methods). That 40% or 3.08 mm y$^{-1}$ offset is three times higher than what was estimated for reefs in south Florida[22], likely because of differences in methods for estimating reef-accretion potential. Although this result supports the conclusion that reef-accretion potential almost certainly overestimates millennial-scale reef accretion[16,22,48], geological estimates of reef accretion, based on limited dating of reef-core records, also necessarily underestimate reef accretion over the shorter, multi-decadal timescales that are most relevant for coral-reef management. Given both the uncertainties in reef-accretion potential and in the overall efficacy of large-scale restoration amid a regime of ongoing disturbance, it is likely that the most realistic outcome will likely lie somewhere in between these two end members.

Importantly, in our projections, the possibility of even a 40% reduction in realized reef accretion does not change our overall conclusion that if restoration can produce a sustained increase of 30% cover of *A. palmata* (≥36% total cover) on the BIRNM reef crest, then the reef could have the potential to keep pace with the low-end sea-level projection (~0.36 m) expected with moderate emissions reductions[31]. Under more pessimistic sea-level-rise scenarios, following impacts of major pulse disturbances like storms, or if the impacts of climate change on coral reefs increase rather than abate over time, then additional restoration efforts would likely be necessary for reefs to provide optimal coastal protection. One possibility in this case would be the adoption of a hybrid restoration approach wherein engineered structures could provide a baseline increase in reef elevation upon which coral restoration can be initiated[59].

Our study demonstrates how restoration could help to mitigate reef degradation and coastal flooding in locations like BIRNM. It also supports the conclusion that for restoration to move beyond short-term enhancement of coral populations and achieve a meaningful impact on reefs' long-term accretion function, will require a substantial investment of resources in the near-term[23]. Bayraktarov et al.[60] found that the median cost of restoration projects globally was ~$400,000 in U.S. dollars per hectare (USD ha$^{-1}$), which equates to ~$6.6 million USD to restore an area the size of the reef crest at BIRNM (166,407 m$^2$; Fig. S1). Recent estimates from nearby south Florida as part of NOAA's Mission: Iconic Reefs initiative indicate that the cost of restoration per *A. palmata* fragment is ~$25 USD (https://www.fisheries.noaa.gov/southeast/habitat-conservation/restoring-seven-iconic-reefs-mission-recover-coral-reefs-florida-keys). At that rate, increasing *A. palmata* cover on a reef the size of the one at Buck Island to a level that could significantly impact shoreline protection (+30% or 500,000 outplants) would cost ~$12.5 million USD. By bolstering the ability of reefs to provide coastal protection, the high up-front costs of restoration could be balanced by both the direct value of restoration to local communities and economies (e.g., BIRNM contributes >$2 million USD annually in tourism[27]) and the millions of dollars of damage to coastal infrastructure that could be avoided and the people in vulnerable communities that could be protected from flooding[49]. With limited resources, local restoration efforts could also be strategically placed to protect the most vulnerable human communities and infrastructure, or critical natural, cultural, and historic resources.

The realities of the high cost of coral restoration are often met with pessimism over the prospects for long-term restoration efficacy and return on investment, but the alternative—allowing coral-reef degradation to continue—would also come at a cost. At just ~0.2 m on average, the conservatively low estimates of net erosion predicted by the end of the century in our study and other carbonate-budget assessments from the western Atlantic[16,22,39,40] seem trivial; however, realized erosion rates may already be higher than our study suggests and will likely be higher going forward because of projected increases

in the frequency of severe thermal-stress events and storms, and the impacts of ocean acidification, as climate change accelerates[11,39]. Furthermore, even the relatively low rates of maximum erosion estimated in our study (0.45 m by 2100) could almost double the realized water levels over the reef crest at BIRNM by 2100 under SSP1-2.6 to SSP2-4.5 (-0.49 m) to a level comparable to the highest sea-level rise predicted under SSP5-8.5 (0.96 m[31]). Based on our models, this would result in a projected -0.2 m increase in wave-driven water levels during storms (Fig. 5a). On the other hand, by increasing reef elevation, restoration could keep water depths at the reef crest near the lowest predictions of sea-level rise for the end of the century (i.e., -0.30 m[31]), preventing wave-driven increases in coastal flooding during storms, even under pessimistic emissions-reduction scenarios. Additionally, every area of reef restored not only contributes to positive accretion, but it also protects the reef structure with a cap of living coral, staving off erosion[13]. Restoration can, therefore, work on both sides of the carbonate-budget equation to close the gap between rising sea levels and the reef surface. Put another way, whereas coral restoration has the potential to minimize climate-change impacts, doing nothing to combat erosion would amplify them.

Coastal flooding and erosion due to extreme weather events have significant impacts on coastal communities globally, and those impacts are projected to increase as sea-level rises over the coming century[2,3]. Coastal communities and decision-makers will be faced with increasingly difficult choices when allocating limited resources to address the growing array of urgent climate-change impacts, and large investments in coral restoration would have to be weighed against other priorities. Our study helps to support that cost-benefit analysis by providing new guidance on the potential scale of restoration necessary to revive reef accretion and maintain reefs' capacity for coastal protection. The success of restoration will ultimately depend on whether ongoing coral mortality, and its underlying causes, can be mitigated. Nonetheless, our results support the conclusion that if restoration is successful, there is currently a window of opportunity during which substantial restoration efforts in the near term could help combat the worst projected impacts of sea-level rise and resulting coastal-flooding risk by the end of the century, whereas allowing reef erosion to continue unchecked could counteract progress from emissions reductions. We conclude that restoration has the potential to be more than a stop-gap measure: it can provide an important component of the portfolio of solutions that communities and managers can draw from to support the persistence of coral reefs and their function as coastal barriers into the future.

## Methods
### Carbonate budgets
We calculated the carbonate budgets of 54 stratified-random sites within shallow fore-reef, reef-crest, and back-reef habitats ($n = 18$ sites in each habitat; Fig. S1) on the windward (southern sector) and leeward (northern sector) reefs ($n = 27$ sites in each sector) of BIRNM. For each site, carbonate budgets were estimated by analyzing reef-census data collected in July 2016 (ref. 61), following a modification of Hubbard et al.[37] and the ReefBudget v1 methodology[44].

To estimate carbonate production, we collected photographs of one $10 \times 1$ m benthic transect at each site and conducted point-count analysis (10–13 images per transect, 75 points per image) using the program CoralNet (https://coralnet.ucsd.edu/) to quantify the percent cover of benthic calcifiers (corals and coralline algae) and total consolidated substratum. Gross carbonate production (kg CaCO$_3$ m$^{-2}$ y$^{-1}$) was estimated by calculating the product of percent cover of each calcifying taxon, taxon-specific calcification rates from ReefBudget v1[44], and, for corals, taxon-specific rugosity correction-factors (Table S6). The rugosity term was included to correct for the fact that the percent cover data in this study were planar, rather than the three-dimensional measurements suggested in ReefBudget (cf.[62]).

Carbonate production rates for all calcifying taxa were summed for each transect.

To estimate parrotfish bioerosion, we recorded the number, size classes (in 15-mm bins), and life phases (initial or terminal) for each of seven bioeroding parrotfish species—*Scarus vetula*, *Sc. taeniopterus*, *Sc. iseri*, *Sparisoma viride*, *Sp. aurofrenatum*, *Sp. rubripinne*, and *Sp. chrysopterum*—within two, $30 \times 4$ m belt-transect surveys at each site. We used the species-, size-, and life-phase-specific bioerosion rates and the average coral skeletal density of 1.67 g cm$^{-3}$ provided in ReefBudget v1[44] (Table S7) to calculate individual bioerosion rates for each species of parrotfish (g ind$^{-1}$ y$^{-1}$). Those rates were multiplied by the densities of parrotfish of each species, size class, and life phase observed within each transect, summed within transect, and averaged between the two transects to estimate total parrotfish bioerosion at each site (kg CaCO$_3$ m$^{-2}$ y$^{-1}$).

We conducted surveys of the bioeroding sea-urchin species *Echinometra lucunter*, *Ec. viridis*, *Diadema antillarum*, and *Eucidaris tribuloides* and recorded their test sizes within 20-mm bins along the same $10 \times 1$ m transects used for the benthic surveys to estimate size-specific densities of each species. Sea-urchin bioerosion (kg CaCO$_3$ m$^{-2}$ y$^{-1}$) was calculated by multiplying the urchin densities by size-specific bioerosion rates derived from the generalized relationship between urchin test size and bioerosion rate described in the ReefBudget v1 methodology[44] (Table S8), which we scaled using a correction factor of 0.57 to account for the proportion of sediment reingested during grazing[63]. Those data were summed across species and size classes to estimate total urchin bioerosion at each site.

We estimated site-level endolithic macrobioerosion (e.g., by clionid sponges, mollusks, and polychaete worms) by multiplying the available substratum for endolithic bioerosion, defined here as the percent cover of dead coral substratum from our benthic surveys, by the average macrobioerosion rate of 0.4 kg m$^{-2}$ y$^{-1}$ measured by Whitcher[43] for the reefs at nearby St. John, U.S. Virgin Islands. Similarly, we calculated site-level microbioerosion (bacteria, algae, and fungi) by multiplying the percent available substrate by the microbioerosion rate of 0.27 kg m$^{-2}$ y$^{-1}$ derived by Vogel[64] in the Bahamas (cf. generalized Caribbean microbioerosion rate from ReefBudget v2: 0.24 kg m$^{-2}$ y$^{-1}$ [ref. 65]). Total bioerosion at each site was estimated as the sum of bioerosion by parrotfish, sea urchins, and endolithic macro- and microbioeroders.

Net carbonate production (kg CaCO$_3$ m$^{-2}$ y$^{-1}$) was estimated by subtracting bioerosion from gross carbonate production at each site and was converted to estimates of reef-accretion potential using information on local reef-framework density and porosity[34]. For sites with positive net carbonate production, we estimated reef-framework density as the weighted average of the densities of coral taxa at the site according to the following equation:

$$D_i = \sum_{n=1}^{n} \left( \frac{x_n}{X_i} \times d_n \right) \qquad (1)$$

where $D_i$ is the weighted mean density of coral framework at site $i$ (kg m$^{-3}$), $x_n$ is the percent cover of coral species $n$ at site $i$, $X_i$ is the percent cover of total live coral at site $i$, $d_n$ is the density of coral species $n$ (kg m$^{-3}$). For net erosional sites we used the average Caribbean coral density value of 1670 kg m$^{-3}$. Estimated vertical reef-framework accretion (or erosion) potential was calculated by dividing net carbonate production by estimated framework density. For sites with positive net carbonate production, we also accounted for the contribution of unconsolidated sediments and void space to total reef accretion using data from Holocene reef cores collected by Hubbard et al.[34] at BIRNM according to the following equation:

$$A_i = F_i + \left( F_i \times \frac{sed}{frame} \right) + \left( F_i \times \frac{void}{frame} \right) \qquad (2)$$

where $A_i$ is the reef-accretion potential at site $i$ (mm y$^{-1}$), $F_i$ is the rate of framework accretion at site $i$ (mm y$^{-1}$), *sed* is the mean percentage of linear depth within the cores that was composed of sediment, *void* is the mean percentage of linear depth within the cores composed of void space, and *frame* is the mean percentage of linear depth within the cores composed of in-place coral framework and rubble. Based on Hubbard et al.[34] *sed* was set at 33%, *void* was set at 18%, and *frame* was set at 49%. The assumed rate of sediment reincorporation at our sites was validated by Whitcher[66] who independently quantified the rate of sediment production from bioerosion at BIRNM. We emphasize that reef-accretion *potential* quantified in our study is likely a maximum estimate of realized reef accretion, as carbonate budget models do not incorporate the impacts of physical erosion or chemical dissolution[16,48].

The ReefBudget v1 methodology[44] has been revised (ReefBudget v2; ref. 65) since this study was conducted, and the updated method would produce some small changes in our estimated budgets; however, because the hydrodynamic models were run using bathymetries created using the budgets we calculated with ReefBudget v1, we did not update the budget calculations here. One of the most substantial changes to the method is that ReefBudget v2 no longer considers bioerosion by juvenile (<10 cm fork length) parrotfish and incorporates substantially lower bioerosion rates for *Sp. aurofrenatum*, *Sp. rubripinne*, and *Sp. chrysopterum*, as these fish may not contribute substantially to bioerosion[65]. It is possible, therefore, that we overestimated bioerosion by those taxa, but because they were substantially less common in our surveys than the more important bioeroding parrotfish, *Sp. viride*, their contribution to overall rates of parrotfish bioerosion in our study was relatively minor (Table S3). For carbonate production, the largest change in ReefBudget v2[44,65] for our study is the increase in the suggested calcification rate for *A. palmata* from 10.88 kg m$^{-2}$ y$^{-1}$ to 14.49 kg m$^{-2}$ y$^{-1}$, which would equate to rates of 36.24 kg m$^{-2}$ y$^{-1}$ and 48.24 kg m$^{-2}$ y$^{-1}$ after incorporating the rugosity-correction factor we used for this species (Table S6). The fact that the highest calcification rate we measured for *A. palmata* at BIRNM was 42.91 kg m$^{-2}$ y$^{-1}$, suggests that the relatively lower calcification rate for this key reef builder suggested in ReefBudget v1 is more appropriate for BIRNM. Overall, the relatively lower parrotfish bioerosion rates and the relatively higher carbonate production rates for *A. palmata* in ReefBudget v2 would produce higher estimates of reef-accretion potential. Thus, the carbonate budgets we derived in this study using ReefBudget v1 may be conservatively low. Similarly, although we also measured local *A. palmata* and *Ps. strigosa* calcification rates at BIRNM (next section), those data were collected after the carbonate budget and hydrodynamic modeling for this study was completed, so we were not able to incorporate them into the 2016 carbonate-budget models; however, the range of local calcification rates (see Results) was similar to the regional estimates we derived from ReefBudget v1[44], so the use of regional rather than local calcification data is likely not a significant source of uncertainty in the models.

All statistical analyses of the were conducted in R Studio (v 4.0.4; R Core Team[67]). We statistically evaluated spatial variability in coral cover, gross carbonate production, bioerosion, and coral reef-accretion potential at BIRNM with linear mixed-effects models (LMEs) using the *nlme* package[68]. Reef sector (northern and southern) and habitat (fore reef, reef crest, and back reef) were treated as fixed factors and site was treated as a random intercept in the models. To further evaluate the drivers of differences among zones (sector and habitat), we compared the contributions of calcifying taxa to carbonate production using ANOSIM and SIMPER in the *vegan* package[69]. We summarize the results of the LMEs using the *anova* function and evaluate significant fixed effects using the Tukey method in the *emmeans* package[70].

To predict how coral-reef degradation will change the bathymetry of the BIRNM fringing reef in the future, we calculated the estimated change in reef elevation for each site by 2100 (i.e., 84 years from when our surveys were conducted in 2016) based on estimated reef-accretion potential. We modified the 2014 USGS lidar bathymetry for BIRNM (in ArcGIS v.10.5.1) based on the mean projection for each habitat in each sector, using benthic habitat maps created by the National Oceanic and Atmospheric Administration (NOAA; https://products.coastalscience.noaa.gov/collections/benthic/e93stcroix/) to delineate the spatial footprint of fore-reef, reef-crest, and back-reef habitats. The 2100 bathymetry based on estimated reef-elevation change was used to estimate the impact of reef degradation on future total water levels at BIRNM (see Hydrodynamic Modeling section).

## Potential impacts of restoring reef-building corals

To assess the potential for restoring populations of reef-building corals and, therefore, reef-accretion potential at BIRNM, we measured the vertical growth and calcification rates of *A. palmata* and *Ps. strigosa* at three locations around the island from June 2019 to July 2021[30], following the methods described in Morrison et al.[71] and Kuffner et al.[19]. Five "mother" colonies (purportedly unique genetic clones) of *A. palmata* were selected within BIRNM by Z.H.S. for subsampling for our study based on knowledge of each colony's history with respect to survival during disease outbreaks and bleaching events, or recruitment to environments of interest to park resource managers. Thirty *Ps. strigosa* colonies were selected based on target size and depth from the area surrounding each calcification-assessment site (Fig. 1a) except for Northwest Reef, where few colonies were found, so colonies were instead collected from the southern fore reef. Corals (~10-cm-diameter colonies for *Ps. strigosa* and ~6-cm branch tips for *A. palmata*) were transplanted onto PVC discs fitted with stainless-steel bolts that slide through a hole in the top of cement blocks secured to the reef[71]. One specimen of each species was deployed together on each block and the corals were photographed, measured (height and planar dimensions), and weighed using the buoyant-weight technique[71] twice per year for two years. Planar area occupied by each colony was calculated using calipers to measure maximum and minimum diameter of the footprint estimated as an ellipse and used to normalize coral calcification rates.

Because *A. palmata* is the primary species responsible for constructing shallow reef-crest habitats in the western Atlantic[13,18] and is, therefore, the species for which restoration could have the most significant impact on coastal-hazards protection, we assessed how the restoration of this species in particular could impact the potential of reefs at BIRNM to keep pace with future sea-level rise. To do this, we first used the mean (±SE) rate of change in *A. palmata* surface area measured at BIRNM in this study[30] to develop a simple matrix model projecting the potential change in percent cover of *A. palmata* in reef-crest habitats of BIRNM for each decade from 2020 to 2100 following outplanting of 100,000, 250,000, and 500,000 colonies: equivalent to -0.6, 1.5, and 3.0 individuals m$^{-2}$ across the 166,407 m$^2$ total area of the BIRNM reef-crest habitat (Fig. S1). For the purposes of the model, we consider an initial "outplant" to be similar in size to the 6-cm branch tips of *A. palmata* we used in our coral-growth experiment. Our population model also necessarily makes the simplifying assumption that growth of *A. palmata* is isometric. The target numbers of *A. palmata* colonies outplanted in our model are not trivial, but they are similar to historical and planned restoration efforts by large-scale restoration programs in the Florida Keys where coral restoration has been underway for more than a decade[72,73]. The outplanting in our model would occur during a theoretical restoration window from 2020–2030, and we included an initial decade of colony growth (and mortality) from 2030–2040.

Restoration of *A. palmata* has only begun in earnest in the last few years and, as a result, few data are available on the long-term survival of outplanted colonies of this species; however, a study by Garrison and Ward[74] that tracked the fate of transplanted storm-generated *A.*

*palmata* fragments in St. John, U.S. Virgin Islands, found that after 12 years (1999–2011), survival was 3%. Although this rate may seem low, it is similar to the benchmark annual survival rate of 70% proposed by Schopmeyer et al.[72] for the congener *A. cervicornis*, when projected over a decadal scale (i.e., 70% per year translates to 2.82% survival after a decade). Some restoration programs have achieved much higher rates of survival (i.e., 89–99% in Belize after a decade[75]); however, we chose to adopt the relatively conservative estimate of 3% for our model.

Low population sizes have significantly limited sexual reproduction of *A. palmata* in recent decades[52], which makes accurately parameterizing reproductive rates problematic. Because the inclusion of the presently low rates of sexual reproduction by *A. palmata* is not likely to significantly change its population projections[56], we did not include sexual reproduction in our model. We did, however, account for the high rate of asexual fragmentation in this species, using the generalized (i.e., not size-specific) annual fragmentation rate of 10% and fragment-mortality rate of 40% for *A. palmata* suggested by Lirman[58] projected over a decade to produce a decadal net fragmentation rate of 55%. Those rates are comparable to rates found in a similar study by Vardi et al.[56]. We note that storms would result in a higher net fragmentation rate, which could be either beneficial or detrimental to *A. palmata* populations depending on their frequency[58]. Because of the uncertainties in future storm climates[11], we did not explicitly incorporate storm-related fragmentation into our model; however, the potential for increasing storm impacts is somewhat accounted for in our increasing mortality scenario described below.

Change in population size in our model was parameterized by combining the 3% survival rate suggested by Garrison and Ward[74] with the 55% fragmentation rate[58] to produce population-level survival rate of 58%, which we round to 60% for simplicity. We ran the model with three theoretical scenarios of how mortality could change over time: 1) a fixed, 60% population survival rate over time; 2) increased climate-change impacts over time for which mortality was increased by 5% each decade (44% net survival by 2100); and 3) decreased climate-change impacts over time through increased resilience, acclimatization, and/or adaptation of *A. palmata* for which mortality was decreased by 5% each decade (80% net survival by 2100). We acknowledge that these scenarios present a simplified version of reality because of the uncertainties in the most likely carbon-emission scenarios and in the capacity of corals to acclimate/adapt to long-term climate impacts. Our aim was to encompass a large range of possible mortality rates (20–56%) in our models, while accounting for fragmentation as an important part of the life history of *A. palamta*[58]. We converted projected surface area from the models to percent cover, by dividing the sum of colony areas by the total area of reef-crest habitat on BIRNM of 166,407 m² based on the assumptions that (1) all *A. palmata* remained within the bounds of that habitat area, and (2) that there was no overlap among colonies.

We also re-ran the carbonate budgets to estimate reef-accretion potential using eight theoretical restoration scenarios wherein the cover of *A. palmata* was increased by 5 to 30% at +5% intervals. We consider an increase of 30% to be a conservative ecological limit for our scenarios based on the observation that cover of *A. palmata* on the reef crest at BIRNM was >50% historically[29] and maximum *A. palmata* cover at our sites is ~20% at present. For this analysis, we used the mean calcification rate of *A. palmata* of 29.07 kg m⁻² y⁻¹ measured in this study to provide a more conservative estimate of reef-accretion potential than the value of 36.24 kg m⁻² y⁻¹ we used in the baseline carbonate-budget model.

Reef-accretion potential quantified in our study should be considered an optimistic, maximum estimate of long-term reef accretion[16,48]. Indeed, by comparing empirical, geological records of recent reef accretion to carbonate-budget assessments of reef-accretion potential, Toth et al.[22] suggested that carbonate-budget

models in south Florida may have underestimated net reef erosion by ~1 mm y⁻¹. We used a similar approach in this study by calculating the difference between a carbonate-budget derived estimate of historic reef-accretion potential for the reef-crest environments at BIRNM based on the maximum observed *A. palmata* cover in the 1970s of 62%[35] and our estimates of bioerosion in 2016 and the maximum Holocene reef accretion by *A. palmata* in St. Croix of 13.4 mm y⁻¹ (note that maximum accretion recorded for Caribbean reefs in <5 m water ranges from 10–20 mm y⁻¹; ref. 36). Our estimate of historic carbonate production by *A. palmata* based on that analysis was 16.48 kg CaCO₃ m⁻² y⁻¹, which is remarkably similar to Gladfelter's[35] estimate of *A. palmata* production at BIRNM at the time of 15.18 kg CaCO₃ m⁻² y⁻¹. Finally, whereas these simple, conceptual models are intended to illustrate the potential of restoration to increase coral cover and reef-accretion potential at BIRNM in theory, we acknowledge that important uncertainties remain regarding the success of restoration outcomes in practice.

### Hydrodynamic modeling

XBeach[76,77] is a process-based numerical model that resolves key hydrodynamic processes on coral-reef-lined coasts[5,78,79]. A two-dimensional (2D) XBeach surfbeat model of Buck Island Reef was constructed to simulate wave and water-level transformation across the reef (details provided in[80]). A curvilinear grid was set up with grid resolution varying 4–40 m in cross-shore direction, and alongshore resolution ranging 9–50 m. The 2014 lidar-derived topo-bathymetry of St. Croix, U.S. Virgin Islands provided the model bed levels. To account for the effect of coral-reef roughness on waves and currents, spatially varying hydrodynamic roughness was included in the model following the methodology by Storlazzi et al.[81]. Modeling of hydrodynamics and the resulting different water-level components and total water levels followed the methodology of Quataert et al.[5].

The XBeach model was validated for a 36-h simulation period during an observed large-wave event on 13 January 2016[82]. Wave boundary conditions were generated by a SWAN[83] model with a regional domain around northern St. Croix, nested in a large-scale domain covering the U.S. and British Virgin Islands. The wave model was forced by ERA5 wave time series and wind fields[84] and validated against four months (July–October 2010) of CDIP (2021) wave-buoy measurements at Christiansted Harbor and Fareham, St. Croix, VI. The modeled mean water levels, sea-swell (5–25 s periods) wave heights ($H_{rms,SS}$), and infragravity (>25 s periods) wave heights ($H_{rms,IG}$) by XBeach agree well with the observations (Fig. S4) across four locations off the northern and southern coast (triangles in Fig. 1).

The validated XBeach model was then forced with a range of sea-level-rise scenarios (+0.0, +0.2, +0.5, +1.2, and +2.0 m; cf.[31]) and wave and storm-surge conditions (10- and 50-year storms). Wave conditions were based on Storlazzi et al.[81] and included a $H_s = 4.01$ m and $T_p = 15.8$ s for the 10-year event (similar to a strong tropical storm or Category-1 hurricane[32]) and $H_s = 4.78$ m and $T_p = 18.7$ s for the 50-year event (equivalent to a Category-5 hurricane[32]); the incident wave direction was from 65° for both return periods. Storm surge levels of +0.37 m and +0.64 m were imposed for the 10- and 50-year event, respectively[85], and added to the MHHW level of 0.40 m for a 1-h simulation period. This same set of forcing conditions were run using modeled bathymetry of the BIRNM fringing reef at 2100 based on our estimated rates of reef accretion (and erosion). For all 20 simulations, maximum total water levels along the shoreline were extracted from the model results to assess the impact of future climate scenarios on the nearshore hydrodynamics at Buck Island.

We also assessed the potential impact of coral restoration on total water levels by subtracting the increase in reef elevation at the reef crest predicted by our carbonate budgets under the +30% *A. palmata* restoration scenario for northern and southern reef sectors from the sea-level-rise scenarios We used linear interpolation to predict total

water levels at the adjusted sea levels for each storm scenario in our model. Because our models do not include scenarios of lower-than-present water depths over the reef crest, in cases where the predicted increase in reef elevation with restoration was greater than sea-level rise, we applied the total water levels for the +0.0 sea-level-rise scenario.

## Data availability

All data used in this study are available in USGS Data Releases (reef-survey data: https://doi.org/10.5066/P97YB2YF; coral-growth data: https://doi.org/10.5066/P94BOI9T; hydrodynamic data: https://doi.org/10.5066/P947RPG4).

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

## Acknowledgements

We are grateful to Elizabeth Whitcher, who led the collection of the reef-census data and generated the initial carbonate budgets as part of a M.S. Thesis at the Florida Institute of Technology and contributed to the manuscript. We also thank Jeff Gay, Robert Fidler, Garrett Fundakowski, and Richard Berey for fieldwork assistance and Erinn Muller for her valuable feedback on the manuscript. This research was supported by an award to IBK and CDS from the U.S. National Park Service's Natural Resource Stewardship and Science program, titled "Mitigate Coral Reef Degradation at Buck Island Reef NM" (PMIS 229486) and funding from the U.S. Geological Survey through the Coastal and Marine Hazards and Resources Program. L.T.T., E.M.W., and R.B.A. were partially funded by the U.S. National Science Foundation (grant OCE-1535007) and the Florida Institute of Technology. The work was performed under scientific collection permit BUIS-2019-SCI-0004 and is study number BUIS-00086. Any use of trade, firm, or product names is for descriptive purposes only and does not imply endorsement by the U.S. Government. This is contribution no. 250 from the Institute for Global Ecology at the Florida Institute of Technology.

## Author contributions

L.T.T., C.D.S., and I.B.K. conceived of the study. I.B.K., A.S., L.T.T., N.H.H., K.A.E., and T.C. collected field data with the support of C.G.P., Z.H.-S., and R.B.A. L.T.T. generated the carbonate budget models and C.D.S., E.Q., J.R., and R.M. L.T.T. prepared the initial draft of the manuscript with C.D.S. and I.B.K. R.B.A. edited the draft, and all authors contributed to the final draft. Retired: Z.H.-S.

## Competing interests

The authors declare no competing interests.
