## [Peer Review File · Nature Communications]

The potential for coral reef restoration to mitigate coastal flooding as sea levels riseEditorial Note: Parts of this Peer Review File have been redacted as indicated to remove third-party material where no permission to publish could be obtained.

REVIEWERS' COMMENTS

Reviewer #1 (Remarks to the Author):

This is the article we need for the times we have! The detailed research emerging out of USGS and partners regarding the wave attenuating benefits of reefs is the most compelling among many compelling reasons to preserve reefs for human life and built capital protection. Restoring reefs to provide this ecosystem service along with the myriad other services reefs provide is the necessary corollary as reef conditions continue to be threatened by multiple anthropogenic sources. I commend the authors for undertaking this critical research and it was an honor to review it.

The authors indicate correctly that combining the work of present coastal flooding inundation with potential for reef accretion via restoration has not been done. This is original work and could help prioritize restoration efforts according to potential for maximum coastal protection.

Several noteworthy results explicated below. This paper will be of MAJOR significant to the field of restoration ecology. This team has done the diligent work not only of measuring and synthesizing the critical variable (SLR, bioerosion, accretion, growth, cost, and mortality of coral fragments ...) but modeling to relevant future dates 2050 and 2100. Even though the benefits from restoration look incremental, perhaps miniscule, the authors are right to assert that the cost of doing nothing is large and the many effects of climate change will exacerbate the accelerating rate of coral reef degradation. The methodology, data analysis, interpretation, presentation, and conclusions are sound. Every time I tried to poke a hole it was addressed upon a closer reading. This work could and should be reproduced in areas that reefs protect significant built capital and human population. Job well done!

- Tali Vardi

ABSTRACT: in line 38 are you generalizing that restoration of *A. palmata* could provide mitigation if restored anywhere, or are you making the point for Buck Island specifically. If you are generalizing you may want to replace IN THE NEAR TERM with IN THE CARIBBEAN. If you are keeping your point narrow - i would add " at this location" or similar.

Figure 1B. Looks like if you took the blue bars of carbonate production and subtracted the red bars, at each location, you would have net carbonate dissolution everywhere. This should be noted and discussed. (Perhaps it is in the text and I haven't gotten there yet!)

Figure 2B. Why "lowest projection of SLR"? Wouldn't it be better to model the minimum *A. palmata* cover needed to keep pace with the worst case scenario? UPDATE I see you addressed the nuances of increased *A. palmata* cover vis-a-vis SLR in lines 228-232. Very nice.

Figure 4. I can't see a difference between A & C. Despite squinting and spending 5 full minutes looking, it seems that the point of the figure is made in panels EF and GH. I'm not sure that panels ABCD are necessary. (Although they are pretty.)

LINE BY LINE:

38 *Acropora palmata* in the n could provide sustained mitigation.

234 - "total water level" is a new term for me that I had to look up. might consider defining it here. Unless I

254-260: This is a really important result. Curious to know what would it take to make the same types of estimates in other locations for which we don't have such detailed bio-erosion and CaCO₃ accretion data. : "Because aggressive coral restoration (i.e., +30% *A. palmata* cover to $\geq 36\%$ total cover) could promote reef growth and help narrow the gap between reef elevation and sea level by more than 0.5 m on average (Fig. 3), restoration could substantially reduce total water levels during storms (Fig. 5B). Restoration has the potential to negate the wave-driven component associated with low sea-level rise scenarios (i.e., by keeping pace with rise of 0.2–0.5 m by 2100)

and to substantially reduce total water levels, by up to 0.81 m (± 0.03) on average, under intermediate-to-high sea-level rise scenarios (i.e., 1.2–2.0 m by 2100). Furthermore, by mitigating wave-driven increases in total water levels, successful, large-scale restoration could essentially reduce the flooding impacts of a major hurricane (50-year storm) to that equivalent to a strong tropical storm or minor hurricane (10-year storm; Fig. 5B).

373 - I see what you mean here that *A. palmata* colonies keep pace with even the most pessimistic SLR scenarios predicted by 2100 SLR. Though there is a slowing down of that pace from 2050 to 2100.

377-380 - I'm glad you are making this point. One important counter point that needs to be addressed somewhere. (Perhaps here.) Is how many things might kill these corals, what killed them in the first place, and why it's worth it to do it anyway. I recognize that all the various 'killers' are wrapped up in the mortality rates - but it's good to be explicit about knowing the many causes of death (as you do in the intro) and the potential that it may still be worth it to engage in large scale restoration in an experimental fashion.

412 - could cite Vardi et al here or at 415 too if you like. No pressure!

454 - stratified instead of strategized?

475-477 - Well put!

640 & 662 - If you'd like to reference a paper that also justified not including sexual reproduction in a matrix population model and presenting the results of # of colonies as percent cover, you can reference Vardi et al 2012.

641 Annual fragmentation rate is size dependent, so not sure what a rate of 10% means.

Reviewer #2 (Remarks to the Author):

Review comments – Toth et al.

This is a very interesting paper by Toth et al and as one would expect from this authorship it is well written and nicely illustrated. The theme of the paper is important and speaks to the potential role of restoration to alleviate the worst excesses of sea level rise in relation to the effectiveness of coral reefs as buffers from wave exposure. It integrates a complex methodology involving in-situ assessments with coastal wave modelling and projections of restoration effectiveness. The outcomes of restoration are I think really best guesses, but the wider aims here are useful in terms of contributing to debate about restoration benefits - although of course key to all this is really the need to remove the causes of initial decline e.g., in this region disease events etc.

Sadly, I think what this study really shows is that given low end SLR scenarios are now almost impossible to achieve a key message here that needs to be clearer is that whilst restoration may help (if external drivers are removed) those restoration efforts are actually most important to limit rather than remove the impacts of realistic future SLR rates. The abstract at present tends to imply that restoration will solve the problem of increasing water depths above reefs. Sadly I don't think it will, but it could help limit the worst outcomes. Fig 3 shows this very clearly that even with very effective restoration and high survival water depths will still increase under what most people would consider more realistic SLR scenarios. This is not to negate the value of the paper and indeed I felt that the realities and challenges and nuances of using restoration and the cost-benefits actually all start to come out very nicely at the end of the discussion. My main question here is does the abstract do a good job of making this clear.

Suggestion: Many people will only read the abstract and use that to remember the key message. I'm not sure it does this as well as it could because it is more a description of what the paper does not the key outcomes – those being that very effective restoration if combined with protection

from external stressors may largely mitigate impacts of low end SLR, but under more realistic mod-to high end SLR these actions whilst not allowing reefs to keep pace are still really important because they will reduce worst excesses of SLR. A key recommendation is thus to revisit the abstract – refer to projected accretion rates and provide a more realistic appraisal of what restoration could help achieve. The title may need a revisit as a result - perhaps almost posed more as a question "What potential for ... "

On the methods I noted that they used methods and data in ReefBudget v1 but also stated "The ReefBudget v1 methodology has been revised (ReefBudget v2) since this study was conducted, and the updated method would produce some changes in our estimated budgets". There follows some discussion of why the data were not updated to make use of the ReefBudget updates which the website suggest were made in 2019.

Suggestion: Given some inevitable differences it would be useful to know, even if run for only a few sites, what the magnitude of difference in production and erosion rates are between the old and new version of this method. I feel it would be useful to test this and support this methods discussion with a few examples to support their statement that the impacts of not updating their findings to use the newer version are valid.

A question on the selected SLR reference points

Line 207 "global mean sea level-rise scenarios outlined in Sweet et al.. ". Why are global mean SLR scenarios used here and not regional ones given that region to global projections can be very different? Please justify. A rate 1 mm per year higher will aggregate to possibly quite relevant to wave exposure models.

In terms of the key questions asked by Nat comms of reviewers:

- What are the noteworthy results? Yes, see my opening comments above.
- Will the work be of significance to the field and related fields? How does it compare to the established literature? If the work is not original, please provide relevant references/ again – see opening comments. I feel the work is important given that this is a topic of great socio-economic relevance.
- Does the work support the conclusions and claims, or is additional evidence needed? See my review comments but I do feel especially that the real relevance of this lies less in suggesting restoration is an answer to all our problems in this area, but that if done well and external factors can be mitigated (both big ifs) that restoration should help limit the worst excesses of SLR. I would suggest that what would be very useful would be to make clear in the abstract what magnitude of SLR we might expect by 2050 and 2100 if restoration effective, what they would mean in projected net water depth increase, and what effect that would have on wave energy reaching the coast i.e., would it still reduce potential exposure by 25%, 50% etc ... ?
- Are there any flaws in the data analysis, interpretation and conclusions? - Do these prohibit publication or require revision? See my comments on the impact of using more recent carbonate budget methodology – and the benefits of testing the outcomes of this on the data.
- Is the methodology sound? Does the work meet the expected standards in your field? Yes, but also see response to previous comment.
- Is there enough detail provided in the methods for the work to be reproduced? Yes well explained.

Other comments:

L37 - aggressive restoration – seems an odd phrasing to me – suggest to reword here and elsewhere

L40 – removal and management of local stressors also fundamental to ensuring any restoration may be effective.

L44 – not really a barrier (that implies something impermeable I think) but a structure that influences wave exposure. Suggest reword

L62 – "reef-crest habitats that once reached sea level are now significantly deeper in many locations". Is there really widespread evidence of this? What is a significant increase in depth here

re: wave attenuation (cm's – m's?). I am not sure I can think of studies that have really shown and quantified how much water depth has increased on reefs across the region, except one example cited from Florida (would be great to know but I am not sure what the elevation benchmark would really be). Suggest to reword this section.

Line 86 and elsewhere – the terms “reef growth” and “reef accretion” are used a bit interchangeably (would suggest to use one and define the term at the start). I would think vertical accretion might be the better option.

L166 “The model indicates that a single large-scale restoration effort now could allow reefs at BIRNM to maintain high levels of coral cover until around 2050; however, additional local management, global reduction of carbon emissions, and/or the recovery of ecological processes like sexual reproduction (illustrated conceptually by the gray arrow) or a decrease in climate-related impact through acclimatization and/or increased resilience (dotted lines) would be necessary to maintain stable coral populations in the long term”. I very much agree with this but it leads me to really conclude it is a very optimistic scenario indeed given lack of much progress on mitigating many of these external drivers. I suggest this highly optimistic outcome is emphasised a bit more clearly. It is good to have a goal but a lot needs to happen to make it viable and to ensure corals survive and increase in number.

L171 – “B) denotes the minimum increase in *A. palmata* cover (25%, to $\geq 31\%$ total cover) our carbonate budgets suggest is needed for reef accretion potential to keep pace with lowest projections of sea-level rise for 2100 (Fig. 3).” Again, this would be great but seems optimistic given points above. I would suggest that this level of optimism needs to be clear in the abstract (what a lot of people will read) because whilst effective restoration may work a LOT needs to happen to make even the minimum restoration outcomes viable.

Coral transplants – how realistic are short term rates compared to longer term – there is good evidence that small corals grow much faster – see work of Carlot et al. Consideration of changes in allometric to isometric growth are relevant here.

L179 – a related point here “based on planar growth rates of *A. palmata* from our study” – how realistic are the growth rates of these small colonies compared to more established colonies. Rates probably reduce once colonies establish – see Carlot et al. It might not be possible to factor for this but a note of consideration would be useful.

L181 onwards – AP growth and cover is important here but actually for reef accretion (the vertical change in reef surface elevation or framework accumulation) it is really the amount of broken framework produced and supplied to the surface of the reef (and retained) that is important. Given the inevitable increase in water depths one may assume higher removal rates of whatever is produced? I think this needs some explanation/consideration.

L372 “our results suggest that growth of individual outplanted *A. palmata* colonies could keep pace with even the most pessimistic sea-level rise scenarios predicted by 2100 (Fig. 3). Although vertical coral growth is not the same as reef accretion, restoration of a barrier of *A. palmata* colonies on the reef crest could have the near-term effect of increasing wave breaking across the reef crest.” I feel that this needs to be changed – as you say in the second sentence here coral growth is not the same as reef accretion – there is as many older reviews have shown (e.g., Dullo) an about order of magnitude difference on average between the two, so I think even speculating here is not helpful. I would delete this section.

L398 “those projected by 2100 under intermediate-to-high CO₂ emissions scenarios (i.e., shared socioeconomic pathway [SSP] 8.5).” I would suggest you make clear that the widespread view is that these intermediate SLR rates are the minimum we are probably going to have to deal with. Low rate rises are now very unlikely in the medium term because of the inherent lag time between ocean warming and expansion. It would be better to say “were low SLR scenarios achievable effective management or restoration might work, but the reality is that rates are almost certainly going to be higher and so the best effects of restoration will be to limit worst excesses of SLR” – that is still super important I think and more realistic.

Reviewer #3 (Remarks to the Author):

This paper provides an important study of the role of reef erosion and accretion, SLR, and the potential effects of large scale restoration with elkhorn coral on the total water levels affecting the coastline of Buck island. While many previous studies of coastal protection with elkhorn coral assume large (and probably unrealistic) changes in water depth associated with restoration, this study provides a more rigorous approach of the role of restoration with elkhorn coral on overall reef accretion and upkeep with SLR, and the resulting benefits in wave attenuation.

The paper is very well written and provides some very important conclusions that should provide useful for proponents of restoration with elkhorn coral. While the paper narrative is excellent, the figures and results could have been more clear and could have shown the results more convincingly and perhaps done justice to the level of work conducted by the authors. However, I think the paper is worthy of being published in Nature Communications. Below are some specific comments:

Figure 4: It is difficult to understand the very significant variability in the difference in total water levels along the shoreline. Why is the distribution almost discontinuous? There are large sections of the coast in which the change is almost zero for all SLR scenarios, but there are some very specific locations where there is a large difference. Is it because of local reef characteristics? Concave vs convex coastline areas? Areas with more or less bioerosion or other mechanisms?

Supplementary Information

I would strongly suggest providing a map of the XBEACH grid and snapshots of the significant wave height and contours of water level, including the spatial distribution in differences for these quantities for a sample wave event. This would help readers interested in the hydrodynamic modeling to better understand the results and perhaps inspire similar studies in other locations. Without these visualizations it is very difficult to understand some of the modeling.

Also, a map of the "restored area" would have been very useful to perhaps better understand this spatial variability. Where exactly in the 2D domain were the colonies out-planted in the model?

As a result of the above, and directly answering the question "Is there enough detail provided in the methods for the work to be reproduced?", I would say that, as submitted, my answer would be NO. This could be easily improved by expanding somewhat on the methodology with better visualizations of the site-specific modeling work (the overall methodology is covered in other cited papers) in the Supplementary Information file.

We would like to thank the three reviewers for their favorable assessment of our manuscript and the constructive suggestions. Our specific response to each of the reviewer's comments are provided in ***bold italics*** below and excerpts from the revised manuscript are *italicized*

Sincerely,

Lauren T. Toth
Corresponding Author

Reviewer #1 (Remarks to the Author):

- 1) This is the article we need for the times we have! The detailed research emerging out of USGS and partners regarding the wave attenuating benefits of reefs is the most compelling among many compelling reasons to preserve reefs for human life and built capital protection. Restoring reefs to provide this ecosystem service along with the myriad other services reefs provide is the necessary corollary as reef conditions continue to be threatened by multiple anthropogenic sources. I commend the authors for undertaking this critical research and it was an honor to review it.

The authors indicate correctly that combining the work of present coastal flooding inundation with potential for reef accretion via restoration has not been done. This is original work and could help prioritize restoration efforts according to potential for maximum coastal protection.

Several noteworthy results explicated below. This paper will be of MAJOR significant to the field of restoration ecology. This team has done the diligent work not only of measuring and synthesizing the critical variable (SLR, bioerosion, accretion, growth, cost, and mortality of coral fragments ...) but modeling to relevant future dates 2050 and 2100. Even though the benefits from restoration look incremental, perhaps miniscule, the authors are right to assert that the cost of doing nothing is large and the many effects of climate change will exacerbate the accelerating rate of coral reef degradation. The methodology, data analysis, interpretation, presentation, and conclusions are sound. Every time I tried to poke a hole it was addressed upon a closer reading. This work could and should be reproduced in areas that reefs protect significant built capital and human population. Job well done!

- Tali Vardi

We thank the reviewer for her favorable assessment and are very pleased that she recognizes the importance of our work.

- 2) ABSTRACT: in line 38 are you generalizing that restoration of *A. palmata* could provide mitigation if restored anywhere, or are you making the point for Buck Island specifically. If you are generalizing you may want to replace IN THE NEAR TERM with IN THE CARIBBEAN. If you are keeping your point narrow - i would add “ at this location” or similar.

Our study provides an example of how successful coral restoration of a ubiquitous Caribbean reef-building coral could help mitigate future coastal flooding. Whereas the exact projections presented in the manuscript are specific to Buck Island, the overall conclusions are potentially broadly applicable in the Caribbean region. Therefore, we do not feel that it is necessary to refer to a specific location here. We have made some other revisions to the abstract in response to the comments by Reviewer 3 (see our response to comment 16).

- 3) Figure 1B. Looks like if you took the blue bars of carbonate production and subtracted the red bars, at each location, you would have net carbonate dissolution everywhere. This should be noted and discussed. (Perhaps it is in the text and I haven't gotten there yet!)

The reviewer is correct that, on average, carbonate production is presently negative throughout BIRNM, as indicated by the first sentence of the results:

Based on our reef census in 2016, our carbonate-budget-estimates of reef-accretion potential varied between -5.31 and 6.21 mm y⁻¹ and averaged -1.56 mm y⁻¹ (± 0.27 SE) across 54 sites...

Because of high carbonate production at a handful of sites on the southern reef crest, our carbonate-budget models suggest that production was roughly neutral in those zones:

Only five sites had positive reef-accretion potential and, on average, reef-accretion potential was only positive on the southern reef crest (0.10 mm y⁻¹ \pm 1.20 SE versus \leq -1.41 mm y⁻¹ in all other zones).

- 4) Figure 2B. Why “lowest projection of SLR”? Wouldn't it be better to model the minimum *A. palmata* cover needed to keep pace with the worst case scenario? UPDATE I see you addressed the nuances of increased *A. palmata* cover vis-a-vis SLR in lines 228-232. Very nice.

Thank you. We did consider modeling even more extensive *A. palmata* scenarios; however, we felt it was important that the scenarios be constrained by both what is feasible under current restoration trajectories (i.e., similar to annual outplanting in some locations like Florida) and ecological baselines.

- 5) Figure 4. I can't see a difference between A & C. Despite squinting and spending 5 full minutes looking. it seems that the point of the figure is made in panels EF and GH. I'm not sure that panels ABCD are necessary. (Although they are pretty.)

We agree that the plots are somewhat duplicative; however, we feel that it is important to present one of the figures showing absolute total water level in addition to the difference plot. We condensed the figure to show the total water levels under the

projection of reef erosion (previously panels C & D) and the absolute difference between the model with and without erosion (previously panels E & F):

We also provided a comparison between the absolute projections with and without erosion (previously panels A-D) in the Supplementary Information (Fig. S2).

- 6) 38 *Acropora palmata* in the n could provide sustained mitigation.
We are not sure what the reviewer meant by this comment, but please see our response to comment 2.

- 7) 234 - “total water level” is a new term for me that I had to look up. might consider defining it here. Unless I
We added the following definition here:
Modeled total water levels—which include the influence of tides, waves, and storm surge—increase non-linearly for increasing sea-level rise and more extreme wave conditions (Figs. 4 & 5) because sea-level rise allows for larger waves to propagate across the reefs (Figs. 5A).

- 8) 254-260: This is a really important result. Curious to know what would it take to make the same types of estimates in other locations for which we don’t have such detailed bio-erosion and CaCO₃ accretion data. : “Because aggressive coral restoration (i.e., +30% *A. palmata* cover to $\geq 36\%$ total cover) could promote reef growth and help narrow the gap between reef elevation and sea level by more than 0.5 m on average (Fig. 3), restoration could substantially reduce total water levels during storms (Fig. 5B). Restoration has the potential to negate the wave-driven component associated with low sea-level rise scenarios (i.e., by keeping pace with rise of 0.2–0.5 m by 2100) and to substantially

reduce total water levels, by up to 0.81 m (± 0.03) on average, under intermediate-to-high sea-level rise scenarios (i.e., 1.2–2.0 m by 2100). Furthermore, by mitigating wave-driven increases in total water levels, successful, large-scale restoration could essentially reduce the flooding impacts of a major hurricane (50-year storm) to that equivalent to a strong tropical storm or minor hurricane (10-year storm; Fig. 5B).

This is an excellent question. We are hesitant to speculate about specific impacts of restoration in other locations, but the broad trends and conclusions we present should be generalizable to many Caribbean reefs. We added the following short paragraph after the paragraph in question to make that point and describe how our specific projections might relate to those for the broader western Atlantic:

*The specific projections from our model are only directly applicable to the baseline ecological and geomorphic setting of BIRNM; however, our overall conclusion—that the worst effects of sea-level rise on coastal flooding could be mitigated if restoration of reef-building species like *A. palmata* can return reef-accretion rates to historic baselines—is relevant for coral reefs throughout the western Atlantic. In fact, given relatively high bioerosion rates we estimated for BIRNM, the level of coral cover needed for reef-accretion potential to match projected rates of sea-level rise may be lower in many other western Atlantic locations^{16,38}. On the other hand, reefs that have already fallen behind present day sea level¹⁴ would have a higher deficit to make up. Developing location-specific coral-restoration strategies is essential for quantifying how broader management goals like shoreline protection can be met^{23,53} and our study demonstrates the utility of carbonate-budget models as a starting point for assessing those types of functional metrics.*

- 9) 373 - I see what you mean here that *A. palmata* colonies keep pace with even the most pessimistic SLR scenarios predicted by 2100 SLR. Though there is a slowing down of that pace from 2050 to 2100.

Our point here is simply that the measured rate of coral growth (70 mm y⁻¹) is higher than the maximum rate of sea-level rise predicted until the end of the century (i.e., 13.4 mm y⁻¹ under RCP 8.5). We clarified the text as follows:

*With high survival and average vertical growth rates of ~70 mm y⁻¹ at BIRNM, our results indicate that individual outplanted *A. palmata* colonies could grow as fast as sea-level rise even under the most pessimistic scenarios predicted by 2100 (i.e., up to 13.4 mm y⁻¹ in St. Croix)³¹.*

- 10) 377-380 - I'm glad you are making this point. One important counter point that needs to be addressed somewhere. (Perhaps here.) Is how many things might kill these corals, what killed them in the first place, and why it's worth it to do it anyway. I recognize that all the various 'killers' are wrapped up in the mortality rates - but it's good to be explicit about knowing the many causes of death (as you do in the intro) and the potential that it may still be worth it to engage in large scale restoration in an experimental fashion.

We addressed this suggestion by expanding upon the causes of coral mortality later in paragraph in question. We also make it clear that our discussion here is specific to what would be needed to maintain functional impacts of coral restoration: however, our results also underscore the conclusion that under a regime of continuing coral mortality from disease outbreaks, bleaching, and other anthropogenic impacts, maintaining functional impacts of such one-off restoration activities will likely require some increase in coral resilience, acclimatization, and/or adaptation^{1,2}.

11) 412 - could cite Vardi et al here or at 415 too if you like. No pressure!

We added this citation.

12) 454 - stratified instead of strategized?

We did mean strategized, but we can see how “strategized spatially” might be unclear to the reader. We changed the sentence to:

With limited resources, local restoration efforts could also be strategically placed to protect the most vulnerable human communities and infrastructure, or critical and natural, cultural, and historic resources.

13) 475-477 - Well put!

Thank you!

14) 640 & 662 - If you'd like to reference a paper that also justified not including sexual reproduction in a matrix population model and presenting the results of # of colonies as percent cover, you can reference Vardi et al 2012.

Thank you for this suggestion. This reference is now included.

15) 641 Annual fragmentation rate is size dependent, so not sure what a rate of 10% means.

Our model only considered the generalized rates presented in the model of Lirman (2003) because we did not model colony sizes, rather only total area. We added the following text to make it clear that we did not consider size-specific fragmentation in this study:

We did, however, account for the high rate of asexual fragmentation in this species, using the generalized (i.e., not size-specific) annual fragmentation rate of 10%...

Reviewer #2 (Remarks to the Author):

16) This is a very interesting paper by Toth et al and as one would expect from this authorship it is well written and nicely illustrated. The theme of the paper is important and speaks to the potential role of restoration to alleviate the worst excesses of sea level rise in relation to the effectiveness of coral reefs as buffers from wave exposure. It integrates a complex methodology involving in-situ assessments with coastal wave modelling and projections of restoration effectiveness. The outcomes of restoration are I

think really best guesses, but the wider aims here are useful in terms of contributing to debate about restoration benefits - although of course key to all this is really the need to remove the causes of initial decline e.g., in this region disease events etc.

Sadly, I think what this study really shows is that given low end SLR scenarios are now almost impossible to achieve a key message here that needs to be clearer is that whilst restoration may help (if external drivers are removed) those restoration efforts are actually most important to limit rather than remove the impacts of realistic future SLR rates. The abstract at present tends to imply that restoration will solve the problem of increasing water depths above reefs. Sadly I don't think it will, but it could help limit the worst outcomes. Fig 3 shows this very clearly that even with very effective restoration and high survival water depths will still increase under what most people would consider more realistic SLR scenarios. This is not to negate the value of the paper and indeed I felt that the realities and challenges and nuances of using restoration and the cost-benefits actually all start to come out very nicely at the end of the discussion. My main question here is does the abstract do a good job of making this clear.

Suggestion: Many people will only read the abstract and use that to remember the key message. I'm not sure it does this as well as it could because it is more a description of what the paper does not the key outcomes – those being that very effective restoration if combined with protection from external stressors may largely mitigate impacts of low end SLR, but under more realistic mod-to high end SLR these actions whilst not allowing reefs to keep pace are still really important because they will reduce worst excesses of SLR. A key recommendation is thus to revisit the abstract – refer to projected accretion rates and provide a more realistic appraisal of what restoration could help achieve. The title may need a revisit as a result - perhaps almost posed more as a question "What potential for ... "

We have done our best to incorporate the reviewer's suggestions for additional language to add to the abstract (see also comments 19 and 25); however, the amount of detail we could add was limited by the length restrictions. We feel that our adjustments to the language have adequately addressed the reviewer's primary suggestions to: 1) explicitly state the magnitude of sea-level rise that our analysis projects restored reefs could keep pace with, 2) make it clear that restoration will not fully mitigate the impacts of sea-level rise, but could prevent the worst outcomes, and 3) change the tone to indicate that we are presenting optimistic scenarios of successful restoration. The relevant revisions are underlined in the abstract text below:

The ability of reefs to protect coastlines from storm-driven flooding hinges on their capacity to keep pace with sea-level rise. We evaluated how and whether coral restoration could achieve the often-cited goal of reversing the impacts of coral-reef degradation to preserve this essential function. We combined coral-growth measurements and carbonate-budget assessments of reef-accretion potential at Buck Island Reef, U.S. Virgin Islands, with hydrodynamic modeling to quantify future coastal flooding with and without various coral-restoration scenarios and sea-level rise projections. We provide

*guidance on how restoration of *Acropora palmata*, if successful, could mitigate the most extreme impacts of coastal flooding by reversing projected trajectories of reef erosion and allowing reefs to keep pace with the ~0.5 m of sea-level rise expected by 2100 with moderate carbon-emissions reductions. This highlights the potential long-term benefits of pursuing coral-reef restoration alongside climate-change mitigation to support the persistence of essential coral-reef ecosystem services.*

- 17) On the methods I noted that they used methods and data in ReefBudget v1 but also stated “The ReefBudget v1 methodology has been revised (ReefBudget v2) since this study was conducted, and the updated method would produce some changes in our estimated budgets”. There follows some discussion of why the data were not updated to make use of the ReefBudget updates which the website suggest were made in 2019.

Suggestion: Given some inevitable differences it would be useful to know, even if run for only a few sites, what the magnitude of difference in production and erosion rates are between the old and new version of this method. I feel it would be useful to test this and support this methods discussion with a few examples to support their statement that the impacts of not updating their findings to use the newer version are valid.

We appreciate the reviewer’s concern; however, there is nothing to suggest that there is anything inherently wrong with the ReefBudget v1 method. Indeed, this method provided the foundation for dozens of high-profile publications prior to 2019 when the new version was released. Because the hydrodynamic models were run based on parameters derived from our original carbonate budget calculations, we feel that re-running those models, when those data cannot actually be incorporated into our study would simply confuse the reader. As a compromise, we have expanded the text in this section to further describe how the differences in the carbonate-budget methods would impact our overall results and provide evidence that our method actually provides a more conservative estimate reef-accretion potential than that which would be derived from the updated methodology:

*The ReefBudget v1 methodology³ has been revised (ReefBudget v2; ref.⁴) since this study was conducted, and the updated method would produce some small changes in our estimated budgets; however, because the hydrodynamic models were run using bathymetries created using the budgets we calculated with ReefBudget v1, we did not update the budget calculations here. One of the most substantial changes to the method was the way in which parrotfish bioerosion was estimated. ReefBudget v2 no longer considers bioerosion by juvenile (<10 cm fork length) parrotfish and incorporates substantially lower bioerosion rates for *Sp. aurofrenatum*, *Sp. rubripinne*, and *Sp. chrysopterum*, as these fish may not contribute substantially to bioerosion⁴. It is possible, therefore, that we overestimated bioerosion by those taxa, but because they were substantially less common in our surveys than the more significant bioeroding parrotfish, *Sp. viride*, their contribution to overall rates of parrotfish bioerosion in our study was relatively minor (Table S3). For carbonate production, the largest change in ReefBudget v2^{3,4} for our study is the increase in the suggested calcification rate for *A. palmata* from*

*10.88 kg m⁻² y⁻¹ to 14.49 kg m⁻² y⁻¹, which would equate to rates of 36.24 kg m⁻² y⁻¹ and 48.24 kg m⁻² y⁻¹ after incorporating the rugosity-correction factor we used for this species (Table S6). The fact that the highest calcification rate we measured for *A. palmata* at BIRNM was 42.91 kg m⁻² y⁻¹, suggests that the relatively lower calcification rate for this key reef builder suggested in ReefBudget v1 is more appropriate for our study. Overall, the relatively lower parrotfish bioerosion rates and the relatively higher carbonate production rates for *A. palmata* in ReefBudget v2 would produce higher estimates of reef-accretion potential. Thus, the carbonate budgets we derived in this study using ReefBudget v1 may be conservatively low.*

18) A question on the selected SLR reference points

Line 207 “global mean sea level-rise scenarios outlined in Sweet et al.. “. Why are global mean SLR scenarios used here and not regional ones given that region to global projections can be very different? Please justify. A rate 1 mm per year higher will aggregate to possibly quite relevant to wave exposure models.

We did use the regional sea-level rise projection developed by Sweet et al. and have reworded the sentence in question as follows to make that clearer:

Median projections (solid to dashed lines; shading represents two standard deviation confidence intervals) of future relative sea-level rise for St. Croix (relative to 2016; NOAA Tide Station # 9751401; ref.³¹) to 2050 (A) and 2100 (B) are based on the five (Low to High) sea-level-rise scenarios outlined in Sweet et al.³¹.

19) In terms of the key questions asked by Nat comms of reviewers:

- What are the noteworthy results? Yes, see my opening comments above.
- Will the work be of significance to the field and related fields? How does it compare to the established literature? If the work is not original, please provide relevant references/ again – see opening comments. I feel the work is important given that this is a topic of great socio-economic relevance.
- Does the work support the conclusions and claims, or is additional evidence needed? See my review comments but I do feel especially that the real relevance of this lies less in suggesting restoration is an answer to all our problems in this area, but that if done well and external factors can be mitigated (both big ifs) that restoration should help limit the worst excesses of SLR. I would suggest that what would be very useful would be to make clear in the abstract what magnitude of SLR we might expect by 2050 and 2100 if restoration effective, what they would mean in projected net water depth increase, and what effect that would have on wave energy reaching the coast i.e., would it still reduce potential exposure by 25%, 50% etc ... ?
- Are there any flaws in the data analysis, interpretation and conclusions? - Do these prohibit publication or require revision? See my comments on the impact of using more recent carbonate budget methodology – and the benefits of testing the outcomes of this on the data.
- Is the methodology sound? Does the work meet the expected standards in your field? Yes, but also see response to previous comment.

- Is there enough detail provided in the methods for the work to be reproduced? Yes well explained.

We appreciate the reviewer's favorable assessment of our paper. Regarding the discussion of the sea-level scenarios in the abstract, please see our response to comment 16. Our response about the carbonate budget methods is addressed in comment 17.

Other comments:

20) L37 - aggressive restoration – seems an odd phrasing to me – suggest to reword here and elsewhere

We changed “aggressive” to “successful” here and have replaced “aggressive” elsewhere with similar phrasing.

21) L40 – removal and management of local stressors also fundamental to ensuring any restoration may be effective.

Although we agree with this statement, these factors were not explicitly modeled in our study, and were therefore not included in the abstract because of space limitations.

22) L44 – not really a barrier (that implies something impermeable I think) but a structure that influences wave exposure. Suggest reword

This phrasing is commonly used in the literature (including the studies cited in this sentence) to draw an analogy between the wave-breaking function of reefs and barrier islands, and so we prefer to retain it here.

23) L62 – “reef-crest habitats that once reached sea level are now significantly deeper in many locations”. Is there really widespread evidence of this? What is a significant increase in depth here re: wave attenuation (cm's – m's?). I am not sure I can think of studies that have really shown and quantified how much water depth has increased on reefs across the region, except one example cited from Florida (would be great to know but I am not sure what the elevation benchmark would really be). Suggest to reword this section.

The study we cite here demonstrated that reef elevations have decreased by between several decimeters (Florida) to more than a meter (St. Thomas, USVI, and Maui, HI). At Buck Island, the decrease ranged between 0.2-0.5 m for reef habitats. Thus, we believe it is likely that this phenomenon is indeed widespread, even if changes in elevation have not been quantified in many locations. We did rephrase this sentence to include the observation that reefs throughout the western Atlantic are becoming flatter (which also indicates that mean elevation should be decreasing; Alvarez-Filip et al. 2009):

The unprecedented loss of this ecosystem engineer has fundamentally changed the structure and function of many western Atlantic reefs: reef-crest habitats that once reached sea level are now significantly flatter and deeper in many locations^{14,15}.

- 24) Line 86 and elsewhere – the terms “reef growth” and “reef accretion” are used a bit interchangeably (would suggest to use one and define the term at the start). I would think vertical accretion might be the better option.

As suggested, we changed “reef growth” to “reef accretion” throughout. This sentence now reads:

Here, we first quantify spatial variability in reef-accretion potential, a measure of the maximum capacity for the vertical accretion of a reef...

- 25) L166 “The model indicates that a single large-scale restoration effort now could allow reefs at BIRNM to maintain high levels of coral cover until around 2050; however, additional local management, global reduction of carbon emissions, and/or the recovery of ecological processes like sexual reproduction (illustrated conceptually by the gray arrow) or a decrease in climate-related impact through acclimatization and/or increased resilience (dotted lines) would be necessary to maintain stable coral populations in the long term”. I very much agree with this but it leads me to really conclude it is a very optimistic scenario indeed given lack of much progress on mitigating many of these external drivers. I suggest this highly optimistic outcome is emphasised a bit more clearly. It is good to have a goal but a lot needs to happen to make it viable and to ensure corals survive and increase in number.

We agree that a lot of progress still needs to be made to realize the more optimistic scenarios we modeled and we have done our best to be very clear about those caveats in the manuscript. In response to this comment, we have rephrased the text in several places to make this point more directly. For example:

In the Abstract:

*We provide guidance on how restoration of *Acropora palmata*, if successful, could mitigate the most extreme impacts of coastal flooding by reversing projected trajectories of reef erosion and allowing reefs to keep pace with the ~0.5 m of sea-level rise expected by 2100 with moderate carbon-emissions reductions.*

In the Results:

*Overall, the model indicates that by the end of the century, the effects of restoration in enhancing *A. palmata* cover would largely be reversed under most of the modeled scenarios (i.e., <10% net increase in coral cover); however, in more optimistic scenarios (i.e., fixed-mortality with 500,000 initial outplants and reduced-mortality with 250,000 initial outplants) *A. palmata* cover is projected to remain elevated by more than 10% to 2100.*

*In the most optimistic scenario, a sustained increase of 30% *A. palmata* cover above the 2016 baselines of 6.14% (± 1.40) and 14.86% (± 4.03) total coral cover (0.01 \pm 0.01 and 4.42 \pm 2.37% *A. palmata* cover) in the northern and southern sectors of the BIRNM reef crest, respectively, would bring total coral cover in those zones to ~36 and 45%.*

In the Discussion:

Our models indicate that, if successful, a significant effort to restore A. palmata populations at BIRNM in the near-term (1.5–3.0 individuals m⁻² by 2030) could result in substantial and sustained increases in coral cover...

In the concluding paragraph of the Discussion:

The success of restoration will ultimately depend on whether ongoing coral mortality, and its underlying causes, can be mitigated.

- 26) L171 –“ B) denotes the minimum increase in A. palmata cover (25%, to $\geq 31\%$ total cover) our carbonate budgets suggest is needed for reef accretion potential to keep pace with lowest projections of sea-level rise for 2100 (Fig. 3).” Again, this would be great but seems optimistic given points above. I would suggest that this level of optimism needs to be clear in the abstract (what a lot of people will read) because whilst effective restoration may work a LOT needs to happen to make even the minimum restoration outcomes viable.

Please see our response to comment 16.

- 27) Coral transplants – how realistic are short term rates compared to longer term – there is good evidence that small corals grow much faster – see work of Carlot et al. Consideration of changes in allometric to isometric growth are relevant here.

L179 – a related point here “based on planar growth rates of A. palmata from our study” – how realistic are the growth rates of these small colonies compared to more established colonies. Rates probably reduce once colonies establish – see Carlot et al. It might not be possible to factor for this but a note of consideration would be useful.

We agree with the reviewer that accurately modeling coral growth is challenging, especially for species with complex, branching morphologies. Unfortunately, to our knowledge size-specific growth rates of A. palmata have not been well quantified. Additionally, because we did not model changes in population demography (i.e., size-frequency distributions), but rather only total area occupied by A. palmata, it would not be possible to directly incorporate this information in our model. We did, however, make it clear in this line that we incorporated average planar growth rates from our study and we added the following sentence to the methods to explicitly acknowledge our assumption of isometric growth:

Our population model also necessarily makes the simplifying assumption that the growth of A. palmata is isometric.

- 28) L181 onwards – AP growth and cover is important here but actually for reef accretion (the vertical change in reef surface elevation or framework accumulation) it is really the amount of broken framework produced and supplied to the surface of the reef (and retained) that is important. Given the inevitable increase in water depths one may assume higher removal rates of whatever is produced? I think this needs some explanation/consideration.

This is an interesting question; however, to our knowledge there is no evidence that framework loss consistently increases with depth. In fact, there is evidence that bioerosion may actually decrease with depth (reviewed in Hubbard 2009; <https://doi.org/10.1002/9781444312065.ch1>). Some new evidence suggests that the depth-related pattern of bioerosion may vary among taxa (with higher macrobioerosion in shallow water and higher fish and microbioerosion in deeper water), but the overall pattern of bioerosion with depth remains unclear. Furthermore, hydrodynamic stress on coral rubble would necessarily decrease (not increase) with increasing depth. Overall, most evidence points to the likelihood that framework retention rates should be as high, if not higher, under future water levels. We feel that it is important to keep speculative discussions of this nature out of the Results section of our manuscript; however, we note that we do extensively discuss the issue of framework retention in the Discussion of the uncertainty of our carbonate-budget-based estimates of reef accretion.

- 29) L372 “our results suggest that growth of individual outplanted *A. palmata* colonies could keep pace with even the most pessimistic sea-level rise scenarios predicted by 2100 (Fig. 3). Although vertical coral growth is not the same as reef accretion, restoration of a barrier of *A. palmata* colonies on the reef crest could have the near-term effect of increasing wave breaking across the reef crest.” I feel that this needs to be changed – as you say in the second sentence here coral growth is not the same as reef accretion – there is as many older reviews have shown (e.g., Dullo) an about order of magnitude difference on average between the two, so I think even speculating here is not helpful. I would delete this section.

*Our intent here is not to speculate about what the growth of these coral colonies means for long-term reef accretion, but instead to describe how the new architecture created by living corals could itself contribute to wave breaking (i.e., an additional, short-term impact of restoration of local hydrodynamics). We modified this section as follows: With high survival and average vertical growth rates of ~70 mm y⁻¹ at BIRNM, our results indicate that individual outplanted *A. palmata* colonies could grow as fast as sea-level rise even under the most pessimistic scenarios predicted by 2100 (i.e., up to 13.4 mm y⁻¹ in St. Croix)³¹. By increasing reef complexity and minimizing the divergence between the elevation of the reef crest and sea level, restoration of a living barrier of *A. palmata* colonies could have the near-term effect of increasing wave breaking across the reef crest. This could prevent additional wave-driven increases in water levels, and, therefore, help mitigate coastal flooding⁴⁹. In the longer-term, the ability of restoration to continue to support coastal protection will depend on its impact not just on coral growth, but on the process of reef accretion.*

- 30) L398 “those projected by 2100 under intermediate-to-high CO₂ emissions scenarios (i.e., shared socioeconomic pathway [SSP] 8.5).” I would suggest you make clear that the widespread view is that these intermediate SLR rates are the minimum we are probably going to have to deal with. Low rate rises are now very unlikely in the medium term

because of the inherent lag time between ocean warming and expansion. It would be better to say “were low SLR scenarios achievable effective management or restoration might work, but the reality is that rates are almost certainly going to be higher and so the best effects of restoration will be to limit worst excesses of SLR” – that is still super important I think and more realistic.

The report by Sweet et al. actually suggests that more moderate “Intermediate-Low to Intermediate” magnitudes of sea-level rise (i.e., ~0.5–1 m) are currently the most likely outcome under all the SSP scenarios including SSP 8.5, which predicts 5C of warming:

[Redacted]

We have clarified this point in the Discussion as follows:

*If the most optimistic restoration scenario, in which *A. palmata* cover is increased by ~30% (to $\geq 36\%$ total cover), can be realized, our carbonate-budget models indicate that reef-accretion potential of the reef crest at BIRNM could exceed 5 mm y^{-1} (Fig. 3). That rate is comparable to average regional Holocene baselines of reef accretion³⁶, but lower than the $>10 \text{ mm y}^{-1}$ maximum Holocene accretion rates by *A. palmata*³⁶. It is also too slow for reef elevation change to match the maximum total sea-level change projected by 2100 (i.e., ~2 m in the High scenario³¹), which could occur under the highest CO₂ emissions scenarios³¹ (i.e., shared socioeconomic pathway [SSP]5-8.5). At present, however, the most probable magnitude of end-of-century sea-level rise across all CO₂-*

emission scenarios is between ~0.5 and 1 m (i.e., the Intermediate-Low to Intermediate scenarios in³¹; Fig. 3), with a higher probability of limiting sea-level rise to ~0.5 m if moderate mid-century reductions of CO₂ emissions are achieved (i.e., SSP1-2.6 or SSP2-4.5)³¹. Our models indicate that under that scenario significant (i.e., +30% *A. palmata* cover) and successful restoration could allow the BIRNM reef crest to keep pace with sea-level rise to 2100 (i.e., 0.5 m; Fig. 3). This outcome would reduce total water levels during storms by >10% (0.81 m) under higher sea-level-rise scenarios (Fig. 5) and thereby decrease the potential flooding associated with major (Category-5) hurricanes to levels projected for more minor (Category-1) storms with no mitigation. Given the prediction that climate change will increase the frequency of severe storms¹¹, the ability of coral reefs to mitigate the worst impacts of storms will become increasingly important in the future.

Reviewer #3 (Remarks to the Author):

- 31) This paper provides an important study of the role of reef erosion and accretion, SLR, and the potential effects of large scale restoration with elkhorn coral on the total water levels affecting the coastline of Buck island. While many previous studies of coastal protection with elkhorn coral assume large (and probably unrealistic) changes in water depth associated with restoration, this study provides a more rigorous approach of the role of restoration with elkhorn coral on overall reef accretion and upkeep with SLR, and the resulting benefits in wave attenuation.

The paper is very well written and provides some very important conclusions that should provide useful for proponents of restoration with elkhorn coral. While the paper narrative is excellent, the figures and results could have been more clear and could have shown the results more convincingly and perhaps done justice to the level of work conducted by the authors. However, I think the paper is worthy of being published in Nature Communications. Below are some specific comments:

We thank the reviewer for their favorable assessment of our study and we have addressed each of their specific points below.

- 32) Figure 4: It is difficult to understand the very significant variability in the difference in total water levels along the shoreline. Why is the distribution almost discontinuous? There are large sections of the coast in which the change is almost zero for all SLR scenarios, but there are some very specific locations where there is a large difference. Is it because of local reef characteristics? Concave vs convex coastline areas? Areas with more or less bioerosion or other mechanisms?

The alongshore variability in total water level is driven by preexisting variability in nearshore bathymetry. We added the following sentence to the results to make this clear:

Maximum total water levels vary along the shoreline due to variability in nearshore bathymetry and its subsequent effects on waves and wave-driven water levels;

Supplementary Information

- 33) I would strongly suggest providing a map of the XBEACH grid and snapshots of the significant wave height and contours of water level, including the spatial distribution in differences for these quantities for a sample wave event. This would help readers interested in the hydrodynamic modeling to better understand the results and perhaps inspire similar studies in other locations. Without these visualizations it is very difficult to understand some of the modeling.

Detailed information about the XBEACH models (including XBEACH grid) is available in the USGS Data Release associated with this study. Because of the large size and high resolution of the XBEACH grids, it is not feasible to duplicate that information here, and instead we refer interested readers to the Data Release. We now directly refer the reader to this information at the beginning of the Methods describing the Hydrodynamic Modeling.

- 34) Also, a map of the “restored area” would have been very useful to perhaps better understand this spatial variability. Where exactly in the 2D domain were the colonies out-planted in the model?

We added this figure to the SI, as suggested. It is now Fig S1.

- 35) As a result of the above, and directly answering the question "Is there enough detail provided in the methods for the work to be reproduced?", I would say that, as submitted, my answer would be NO. This could be easily improved by expanding somewhat on the methodology with better visualizations of the site-specific modeling work (the overall methodology is covered in other cited papers) in the Supplementary Information file.

We added the majority of the suggested figures to the Supplementary Information.